# Neural Combinatorial Optimization with Heavy Decoder: Toward Large Scale Generalization

**Fu Luo**[1*]  **Xi Lin**[2*]  **Fei Liu**[2]  **Qingfu Zhang**[2]  **Zhenkun Wang**[1†]

[1] Southern University of Science and Technology
[2] City University of Hong Kong
luof2023@mail.sustech.edu.cn, {xi.lin, fliu36-c}@my.cityu.edu.hk,
qingfu.zhang@cityu.edu.hk, wangzhenkun90@gmail.com

## Abstract

Neural combinatorial optimization (NCO) is a promising learning-based approach for solving challenging combinatorial optimization problems without specialized algorithm design by experts. However, most constructive NCO methods cannot solve problems with large-scale instance sizes, which significantly diminishes their usefulness for real-world applications. In this work, we propose a novel Light Encoder and Heavy Decoder (LEHD) model with a strong generalization ability to address this critical issue. The LEHD model can learn to dynamically capture the relationships between all available nodes of varying sizes, which is beneficial for model generalization to problems of various scales. Moreover, we develop a data-efficient training scheme and a flexible solution construction mechanism for the proposed LEHD model. By training on small-scale problem instances, the LEHD model can generate nearly optimal solutions for the Travelling Salesman Problem (TSP) and the Capacitated Vehicle Routing Problem (CVRP) with up to 1000 nodes, and also generalizes well to solve real-world TSPLib and CVRPLib problems. These results confirm our proposed LEHD model can significantly improve the state-of-the-art performance for constructive NCO. The code is available at https://github.com/CIAM-Group/NCO_code/tree/main/single_objective/LEHD.

## 1 Introduction

Combinatorial optimization (CO) holds significant practical value across various domains, such as vehicle routing [48], production planning [11], and drug discovery [34]. Due to the NP-hardness and the presence of numerous complex variants, solving CO problems remains an extremely challenging task [31]. In the past few decades, many powerful algorithms have been proposed to tackle different CO problems, but they mainly suffer from two major limitations. Firstly, solving each new problem requires extensive domain knowledge to design a proper algorithm by experts, which could lead to a significant development cost. In addition, these algorithms usually require an excessively long execution time due to the NP-hardness of CO problems [49]. These limitations make the classical algorithms undesirable and impractical for many real-world applications.

Recently, many learning-based neural combinatorial optimization (NCO) methods have been proposed to solve CO problems without handcrafted algorithm design. These methods build neural network models to generate the solution for given CO problem instances, which can be trained by supervised learning (SL) [51, 22, 13, 23, 28, 18] or reinforcement learning (RL) [4, 27, 17, 19, 6, 9, 53, 37, 29, 36, 24, 39, 40, 54, 52, 1]. Although these methods can achieve promising performance on

---

*Equal contributors
†Corresponding author

37th Conference on Neural Information Processing Systems (NeurIPS 2023).

small-scale problems, they perform poorly on solving large-scale problems. Due to the NP-hardness of large-scale problems, the SL-based methods struggle to obtain enough high-quality solutions as labeled training data. Meanwhile, the RL-based methods could suffer from the critical issues of sparse rewards [47, 15] and device memory limitations for solving large-scale problems. Therefore, it is impractical to directly train the NCO model on large-scale problem instances. One feasible approach is to train the NCO model on small-scale problems and then generalize it to solve large-scale problems. However, the current purely learning-based constructive NCO methods have a very poor generalization ability, which diminishes their usefulness in solving large-scale real-world problems.

The current constructive NCO models typically have a Heavy Encoder and Light Decoder (HELD) structure, which could be an underlying reason for their poor generalization ability. The HELD-based model aims to learn the embeddings of all nodes via a heavy encoder in one shot and then sequentially construct the solution with the static node embeddings via a light decoder. This one-shot embedding learning may incline the model to learn scale-related features to perform well on small-scale instances. When the model generalizes to large-scale problem instances, the learned scale-related features may hinder the model from capturing the necessary relations among a significantly larger number of nodes. This work proposes a constructive NCO model with a novel Light Encoder and Heavy Decoder (LEHD) structure to tackle this issue. Instead of one-shot embedding learning, our proposed LEHD-based model learns to dynamically capture the relationships among the current partial solution and all the available nodes at each construction step via the heavy decoder. As the size of the available nodes varies with construction steps, the model tends to learn scale-independent features. Moreover, the model can iteratively adjust and refine the learned node relationships during the training. Therefore, it could be less sensitive to the instance size and has much better generalization performance on large-scale problem instances.

It becomes impractical to train the proposed LEHD model with RL due to the huge memory and computational cost of the heavy decoder structure. In this paper, we proposed a data-efficient training scheme, called *learn to construct partial solution*, to train the LEHD model in a supervised manner. With this scheme, the model learns to reconstruct partial solutions during the optimization process, which can be treated as a kind of data augmentation for robust model training. Finally, similar to the training process, we propose a flexible solution construction mechanism called *Random Re-Construct (RRC)* for efficient active improvement at the inference stage, which can significantly improve the solution quality with iterative local reconstructions.

Our contributions can be summarized as follows:

- We propose a novel Light Encoder and Heavy Decoder (LEHD) model for generalizable neural combinatorial optimization. By only training on small-scale problem instances, the proposed LEHD model can achieve robust and promising performance on problems with much larger sizes.

- We develop a data-efficient training scheme and a flexible solution construction mechanism for the proposed LEHD model. The data-efficient training scheme enables LEHD to be trained efficiently via supervised learning, while the solution construction mechanism can continuously improve the solution quality through a customized inference budget.

- Our purely learning-based constructive method can achieve state-of-the-art performance in solving TSP and CVRP at various scales up to size 1000, and also generalizes well to solve real-world TSPLib/CVRPLib problems.

## 2 Related Work

**Constructive NCO with Balanced Encoder-Decoder**    In their seminal work, Vinyals et al. [51] propose the Pointer Network as a neural solver to directly construct solutions for combinatorial optimization problems in an autoregressive manner. This model has a balanced encoder-decoder structure where both the encoder and decoder are recurrent neural networks (RNNs) with the same number of layers. The original Pointer Network is trained by supervised learning, while Bello et al. [4] proposes an efficient reinforcement learning method for NCO model training. Some Pointer Network variants have been proposed to tackle different combinatorial optimization problems [37]. However, these models can only solve small-scale problem instances with sizes up to 100 and have poor generalization performance.

**Constructive NCO with Heavy Encoder and Light Decoder**   Kool et al. [27] and Deudon et al. [10] leverage the Transformer architecture [46] to design more powerful NCO models. The Attention Model (AM) proposed by Kool et al. [27] has a Heavy Encoder and Light Decoder (HELD) structure with three attention layers in the encoder and one layer in the decoder. This model can obtain promising performance on various small-scale problems with no more than 100 nodes. Since then, AM has become a typical model choice, and many AM variants have been proposed for constructive NCO [53, 29, 54, 30, 24]. Among them, POMO proposed by Kwon et al. [29] is an AM model with a more powerful learning and inference strategy based on multiple optimal policies. The POMO model has a heavy encoder with six attention layers and a light one-layer decoder. It can achieve remarkable performance on small-scale problems with up to 100 nodes.

However, these constructive models with heavy encoder and light decoder structures still have a very poor generalization performance on problem instances with larger sizes, even for those with up to 200 nodes [23]. Directly training these models on the large-scale problem is also very difficult if not infeasible [23]. Different methods have been proposed to enhance the generalization ability for constructive NCO models in the inference stage, such as Efficient Active Search (EAS) [19] and Simulation-guided Beam Search (SGBS) [8]. While these methods can improve the constructive NCO model's generalization performance on problems with up to size 200, they struggle to solve problems with larger scales.

Very recently, a few approaches have been proposed to use the AM model to tackle large-scale TSP instances. Pan et al. [38] propose a Lower-Level Model that learns to construct the solution of sub-problems obtained by an Upper-Level Model. Another method proposed by Cheng et al. [7] uses an AM model to learn to reconstruct segments of a given TSP solution and selects the most improving segment to optimize it. However, these methods heavily rely on specialized designs for TSP by human experts, and cannot be used to solve other problems such as CVRP. In a concurrent work, Drakulic et al. [12] propose a novel Bisimulation Quotienting (BQ) method for generalizable neural combinatorial optimization by reformulating the Markov Decision Process (MDP) of the solution construction. Our proposed LEHD model has a different motivation and model structure from BQ, and a comprehensive comparison is provided in the experiment section.

**Non-Constructive NCO Methods**   In addition to constructive NCO, there are also other learning-based approaches that work closely with search and improvement methods [33, 6, 18, 21]. Joshi et al. [22] propose a graph convolutional network to predict the heat map of each edge's probability in the optimal solution for a TSP instance. Afterward, approximate solutions can be obtained via the heat map guided beam search [22], Monte Carlo tree search (MCTS) [13], dynamic programming [28], and guided local search [20]. The heatmap-based approach was originally proposed to solve small-scale problems, and some recent works generalize it for solving large-scale TSP instances [13, 42]. However, they heavily rely on the carefully expert-designed MCTS strategy for TSP [13] and cannot be used to solve other problems.

Learning-augmented methods can also be applied to improve and accelerate heuristic solvers via operation selection [6, 35, 9], guided improvement [17, 5, 55] and problem decomposition [44, 32]. Many improvement-based methods have also been proposed to iteratively improve a feasible initial solution [3, 52, 36]. However, these methods typically rely on human-designed heuristic operators or more advanced solvers [14, 16] for solving different problems. In this work, we mainly focus on the purely learning-based NCO solver without expert knowledge.

## 3   Model Architecture: Light Encoder and Heavy Decoder

In this section, we propose a novel NCO model with the Light Encoder and Heavy Decoder (LEHD) structure to tackle the critical issue regarding large-scale generalization.

### 3.1   Encoder

For a problem instance $\mathbf{S}$ with $n$ node features $(\mathbf{s}_1, \ldots, \mathbf{s}_n)$ (e.g., the coordinates of $n$ cities), a constructive model parameterized by $\boldsymbol{\theta}$ generates the solution in an autoregressive manner, i.e., it constructs the solution by selecting nodes one by one. Our LEHD-based model consists of a light encoder and a heavy decoder, as shown in Figure 1. The light encoder has one attention layer, and the heavy decoder has $L$ attention layers. Given the problem instance with node features

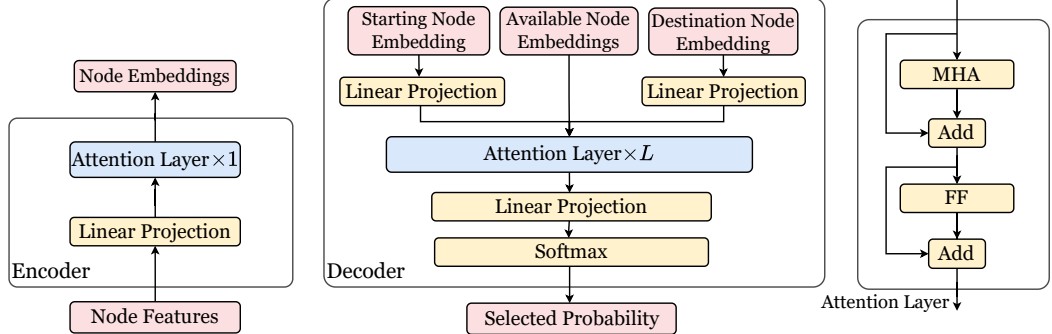

Figure 1: The structure of our proposed LEHD model, which has a single-layer light encoder and a heavy decoder with $L$ attention layers.

$\mathbf{S} = (\mathbf{s}_1, \ldots, \mathbf{s}_n)$, the light encoder transforms each node features $\mathbf{s}_i$ to its initial embedding $\mathbf{h}_i^{(0)}$ through a linear projection. The initial embeddings $\{\mathbf{h}_1^{(0)}, \ldots, \mathbf{h}_n^{(0)}\}$ are then fed into one attention layer to get the node embedding matrix $H^{(1)} = (\mathbf{h}_1^{(1)}, \ldots, \mathbf{h}_n^{(1)})$.

**Attention layer**   The attention layer comprises two sub-layers: the multi-head attention (MHA) sub-layer and the feed-forward (FF) sub-layer [46]. Different from generic NCO models such as AM [27], the normalization is removed from our model to enhance generalization performance (see Appendix A). Let $H^{(l-1)} = (\mathbf{h}_1^{(l-1)}, \ldots, \mathbf{h}_n^{(l-1)})$ be the input of the $l$-th attention layer for $l = 1, \ldots, L$, the output of the attention layer in terms of the $i$-th node is calculated as:

$$
\begin{aligned}
\hat{\mathbf{h}}_i^{(l)} &= \mathbf{h}_i^{(l-1)} + \mathrm{MHA}(\mathbf{h}_i^{(l-1)}, H^{(l-1)}), \\
\mathbf{h}_i^{(l)} &= \hat{\mathbf{h}}_i^{(l)} + \mathrm{FF}(\hat{\mathbf{h}}_i^{(l)}).
\end{aligned}
\tag{1}
$$

We denote this embedding process as $H^{(l)} = \mathrm{AttentionLayer}(H^{(l-1)})$.

## 3.2   Decoder

The decoder sequentially constructs the solution in $n$ steps by selecting node by node. In the $t$-th step for $t \in \{1, \ldots, n\}$, the current partial solution can be written as $(x_1, \ldots, x_{t-1})$, the first selected node's embedding is denoted as $\mathbf{h}_{x_1}^{(1)}$, the embedding of the node selected in the previous step is denoted as $\mathbf{h}_{x_{t-1}}^{(1)}$, and the embeddings of all available nodes is denoted by $H_a = \{\mathbf{h}_i^{(1)} \mid i \in \{1, \ldots, n\} \setminus \{x_1, \ldots, x_{t-1}\}\}$. Since the decoder has $L$ attention layers, the $t$-th construction step of LEHD can be described as:

$$
\begin{aligned}
\widetilde{H}^{(0)} &= \mathrm{Concat}(W_1 \mathbf{h}_{x_1}^{(1)}, W_2 \mathbf{h}_{x_{t-1}}^{(1)}, H_a), \\
\widetilde{H}^{(1)} &= \mathrm{AttentionLayer}(\widetilde{H}^{(0)}), \\
&\quad \cdots \\
\widetilde{H}^{(L)} &= \mathrm{AttentionLayer}(\widetilde{H}^{(L-1)}), \\
u_i &= \begin{cases} W_O \widetilde{\mathbf{h}}_i^{(L)}, & i \neq 1 \text{ or } 2 \\ -\infty, & \text{otherwise} \end{cases}, \\
\mathbf{p}^t &= \mathrm{softmax}(\mathbf{u}),
\end{aligned}
\tag{2}
$$

where $W_1, W_2, W_O$ are learnable matrices. The matrices $W_1$ and $W_2$ are used to re-calculate the embeddings of the starting node (i.e., the node selected in the previous step) and the destination node (i.e., the node selected in the first step), respectively. The matrix $W_O$ is used to transform the relation-denoting embedding matrix $\widetilde{H}^{(L)}$ into a vector $\mathbf{u}$ for the purpose of calculating the selection probability $\mathbf{p}^t$. The most suitable node $x_t$ is selected based on $\mathbf{p}^t$ at each decoding step $t$. Finally, a complete solution $\mathbf{x} = (x_1, \ldots, x_n)^\mathsf{T}$ is constructed by calling the decoder $n$ times.

For the HELD model such as AM [27], the heavy encoder with $L$ layers outputs the node embedding matrix as $H^{(L)} = (\mathbf{h}_1^{(L)}, \ldots, \mathbf{h}_n^{(L)})$. The average of all nodes' embeddings (i.e., $\frac{1}{n} \sum_{i=1}^n \mathbf{h}_i^{(L)}$), the

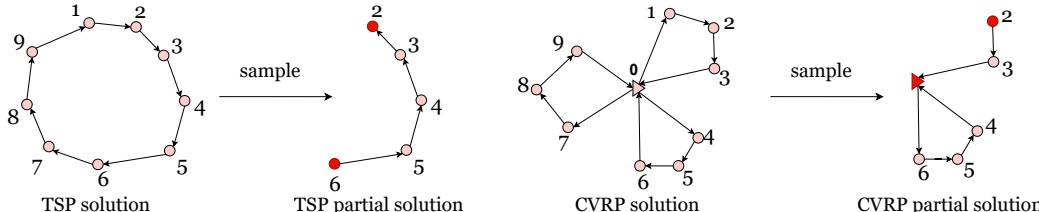

Figure 2: Examples of generating partial solution for TSP and CVRP. For the TSP instance, its optimal solution is [1,2,3,4,5,6,7,8,9], and a partial solution is randomly sampled as [6,5,4,3,2]. For the CVRP instance, its optimal solution is [0,1,2,3,0,4,5,6,0,7,8,9,0], and a partial solution [2,3,0,6,5,4,0] is randomly sampled. We impose a restriction for CVRP that the partial solution must end at the depot.

starting node's embedding (i.e., $\mathbf{h}_{x_{t-1}}^{(L)}$), the destination node's embedding (i.e., $\mathbf{h}_{x_1}^{(L)}$) are contacted to form the context node embedding (i.e., $\mathbf{h}_c^{(0)}$). Thereafter, the light decoder takes the context node embedding and the embeddings of all nodes (i.e., $H^{(L)}$) as input, and computes the selection probability through an MHA operation, the compatibility calculation, and the softmax function, i.e.,

$$\mathbf{h}_c^{(1)} = \text{MHA}(\mathbf{h}_c^{(0)}, H^{(L)}),$$

$$u_i = \begin{cases} 10 \cdot \tanh\left(\dfrac{(W_C \mathbf{h}_c^{(1)})^{\mathsf{T}} W_E \mathbf{h}_i^{(L)}}{d}\right), & i \neq x_{1:t-1} \\ -\infty, & \text{otherwise} \end{cases},$$

$$\mathbf{p}^t = \text{softmax}(\mathbf{u}),$$

(3)

where $W_C$ and $W_E$ are learnable matrices, $d$ is a parameter determined by the embedding dimension. Since the HELD model's decoder only has a static node embedding matrix $H^{(L)}$, the relationships among the nodes it captures are not updated throughout the decoding process. In other words, the HELD model's performance relies heavily on the quality of the static node embedding matrix. When the problem size becomes larger, the ability of the static node embedding matrix to represent the relationships among all nodes becomes worse, resulting in the poor generalization performance of the model.

In our LEHD model, the heavy decoder dynamically re-embeds the embeddings of the starting node, destination node, and available nodes via $L$ attention layers, thus updating the relationships among the nodes at each decoding step. Such a dynamic learning strategy enables the model to adjust and refine its captured relationships between the starting/destination and available nodes. In addition, as the size of the nodes varies during the construction steps, the model tends to learn the scale-independent features. These good properties enable the model to dynamically capture the proper relationships between the nodes, thereby making more informed decisions in the node selection on problem instances of various sizes.

## 4 Learn to Construct Partial Solution

The constructive NCO aims to construct the solution node-by-node using the decoder. The heavy decoder of our LEHD model can result in a higher computational cost than that of the HELD model at each construction step. Whereas the RL training method needs to generate the complete solution before computing the reward, which implies it requires significantly huge memory and computational costs. Furthermore, RL suffers from the challenging issue of sparse reward, especially for problems with large sizes.

According to Yao et al. [56], data augmentation (DA) can significantly reduce the number of required high-quality solutions (i.e., labels) for SL training, and DA-based SL can even outperform RL regarding solution accuracy and generalization performance. Operations based on the property of multiple optimality [29] and optimality invariance [25] are the most commonly used DA methods in NCO. In this work, we develop an efficient DA-based SL training scheme also based on the property of optimality invariance.

For each training data (i.e., a set of nodes), its label is a sequence of all the nodes (i.e., a valid tour). According to the property of optimality invariance, the partial solution of the optimal solution must

also be optimal. Therefore, we can randomly sample labeled partial solutions with various sizes and different directions from each labeled data, thus greatly enriching our training set. Based on these characteristics, we propose a training scheme called *learning to construct partial solutions*. The model learns to construct optimal partial solutions of various sizes and random directions, thereby resulting in a more efficient and robust training process.

### 4.1 Generate Partial Solutions during the Training Phase

For a routing problem instance $\mathbf{S}$ with $n$ nodes $(\mathbf{s}_1, \ldots, \mathbf{s}_n)$, its optimal solution can be represented by a specific node sequence $\mathbf{x}$ (also called the tour). The partial solution $\mathbf{x}_{sub}$ is a consecutive subsequence sampled from $\mathbf{x}$ with a random size and direction. In this work, we let each partial solution have a length with $w$ sampled from the discrete uniform distribution $\mathbf{Unif}([4, |V|])$ where $|V|$ is the number of nodes for each TSP/CVRP instance, since a problem with size less than $4$ is trivial to solve. Each partial solution $\mathbf{x}_{sub}$ is one augmented data point in our method. Two examples of generating partial solutions for TSP and CVRP are provided in Figure 2.

### 4.2 Learn to Construct Partial Solutions via Supervised Learning

The model learns to construct the partial solution node by node. The first node of $\mathbf{x}_{sub}$ is the starting node, the last node of $\mathbf{x}_{sub}$ is the destination node, and the remaining nodes constitute the available nodes to be selected by the model. For the example $\mathbf{x}_{sub} = [6, 3, 5, 4, 1]$, node 6 is the starting node, node 1 is the destination node, and nodes $3, 5, 4$ are available nodes.

In each training step, the model provides each available node $\mathbf{s}_i$ with a probability $p_i$ of being selected. Meanwhile, the label $\mathbf{x}_{sub}$ tells whether the node $\mathbf{s}_i$ should be chosen in the current step (i.e., $y_i = 1$ or 0). The cross-entropy loss $loss = -\sum_{i=1}^{u} y_i \log(p_i)$ is employed to measure the error made by the current model, where $u$ is the number of available nodes. Then we can update the model parameters by a gradient-based algorithm (e.g., Adam) with respect to the loss. Throughout the training process, the number of available nodes gradually decreases, and the starting node dynamically shifts to the node selected in the previous step while the destination node remains constant. The training epoch ends when there are no more nodes available. The next epoch starts with new augmented data.

### 4.3 Generate the Complete Solution during the Inference Phase

During the inference phase, our proposed LEHD model employs a greedy step-by-step approach to construct the whole solution for the given instance $\mathbf{S}$. In the beginning, a single node is randomly selected to serve as the destination node and also the initial starting node. The remaining nodes constitute the set of available nodes that the model can select step-by-step. Once a node is selected, it will be treated as the new starting node and removed from the set of available nodes, i.e., the starting node is dynamically shifting while the destination node remains constant. When no more available nodes exist, the whole solution is generated. In this way, LEHD can easily construct solutions for problems of various sizes as well as unseen large-scale problems.

## 5 Random Re-Construct for Further Improvement

The constructive model has an inductive bias, which means it could construct a better solution by following its preferred direction. Similarly, different starting and destination nodes will lead to different solution quality since the model may perform better when starting from some well-learned local patterns. The tour generated via greedy search could be not optimal and contain multiple suboptimal local segments (e.g., partial solutions). Therefore, rectifying these suboptimal segments during the inference phase can effectively enhance the overall solution quality.

Our model stems from learning to construct partial solutions, making it appropriate for iteratively reconstructing partial solutions from different directions and different starting and destination nodes. It naturally leads to our proposed flexible construction mechanism, *Random Re-Construct*(RRC). Specifically, being the same as generating the partial solution during training, RRC randomly samples a partial solution from the initial solution and restructures it to obtain a new partial solution. If the new partial solution is superior, it replaces the old one. This iterative process can enhance the solution quality within a stipulated time budget.

# 6 Experiment

In this section, we empirically compare our proposed LEHD model with other learning-based and classical solvers on TSP and CVRP instances with various sizes and distributions.

**Problem Setting** We follow the standard data generation procedure of the previous work [27] to generate TSP and CVRP datasets. The training sets consist of one million instances for TSP100 and CVRP100, respectively. The TSP test set includes 10,000 instances with 100 nodes and 128 instances for each of 200, 500, and 1000 nodes, respectively. Similarly, the CVRP test set is constructed with the same number of instances. We obtain optimal solutions for the TSP training set with the Concorde solver [2] and obtain optimal solutions for the CVRP training set with HGS [50]. The training and test datasets can be downloaded from Google Drive [3] or Baidu Cloud [4].

**Model Setting** For our LEHD model, the embedding dimension is set to 128, and the number of attention layers in the decoder is set to 6. In each attention layer, the head number of MHA is set to 8, and the dimension of the feed-forward layer is set to 512. In all experiments, we train and test our LEHD models using a single NVIDIA GeForce RTX 3090 GPU with 24GB memory.

**Training** Both the TSP model and CVRP model are trained on one million instances of size 100, respectively. The optimizer is Adam [26] with an initial learning rate of 1e-4. The value of the learning rate decay is set to 0.97 per epoch for the TSP model and 0.9 per epoch for the CVRP model. With a batch size of 1024, we train the TSP model for 150 epochs and the CVRP model for 40 epochs since it converges much faster.

**Baselines** We compare our method with **(1) Classical Solvers:** Concorde [2], LKH3 [16], HGS [50], and OR-Tools [41]; **(2) Constructive NCO:** POMO [29], MDAM [54], EAS [19], SGBS [8], and BQ [12]; **(3) Heatmap-based Method:** Att-GCN+MCTS [13].

**Metrics and Inference** We report the optimality gap and inference time for all methods. The optimality gap measures the difference between solutions achieved by different learning and non-learning-based methods and the optimal solutions obtained using Concorde for TSP and LKH3 for CVRP. Since the classical solvers are run on a single CPU, their reported inference times are not directly comparable to those of the learning-based methods executed on GPU.

For MDAM, POMO, SGBS, and EAS, we directly run their source code with default settings on our test set. For Att-GCN+MCTS, we report their original results for the corresponding setting. For BQ, following the settings described in the original paper, we reproduce and train the BQ model by ourselves and report the obtained results on our test set. For our proposed LEHD model, we report both the results for greedy inference as well as for Random Re-Construct (RRC) with various computational budgets.

## 6.1 Experimental Results

The main experimental results on uniformly distributed TSP and CVRP instances are reported in Table 1. For TSP, our proposed LEHD method can obtain good greedy inference performance with a very fast inference time on instances of all sizes. With only 100 RRC iterations, LEHD can significantly outperform all other learning-based methods with a reasonable inference time. If we have a larger budget of more than 500 RRC iterations, LEHD can achieve very promising performance for all instances, of which the gap is nearly optimal for TSP100 and TSP200, less than $0.2\%$ for TSP500, and around $0.8\%$ for TSP1000.

For CVRP, LEHD can also achieve promising greedy inference performance for all instances. With 300 RRC iterations, LEHD can outperform most learning-based methods on all CVRP instances, except for the EAS method on CVRP100, which requires a much longer inference time. The EAS's performance significantly decreases for solving larger instances such as CVRP500, and we fail to obtain a result for EAS on CVRP1000 with a reasonable budget. SGBS struggles with poor generalization and long inference time for solving large-scale instances. LEHD can also outperform

---

[3] https://drive.google.com/drive/folders/1LptBUGVxQlCZeWVxmCzUOf9WPlsqOROR?usp=sharing
[4] https://pan.baidu.com/s/12uxjol_5pAlnm0j4F6D_RQ?pwd=rzja

Table 1: Experimental results on TSP and CVRP with uniformly distributed instances. The results of methods with an asterisk (*) are directly obtained from the original paper.

| | | TSP100 | | TSP200 | | TSP500 | | TSP1000 | |
|---|---|---|---|---|---|---|---|---|---|
| Concorde | | 0.000% | 34m | 0.000% | 3m | 0.000% | 32m | 0.000% | 7.8h |
| LKH | | 0.000% | 56m | 0.000% | 4m | 0.000% | 32m | 0.000% | 8.2h |
| OR-Tools | | 2.368% | 11h | 3.618% | 17m | 4.682% | 50m | 4.885% | 10h |
| Att-GCN+MCTS* | | 0.037% | 15m | 0.884% | 2m | 2.536% | 6m | 3.223% | 13m |
| MDAM bs50 | | 0.388% | 21m | 1.996% | 3m | 10.065% | 11m | 20.375% | 44m |
| POMO augx8 | | 0.134% | 1m | 1.533% | 5s | 22.187% | 1m | 40.570% | 8m |
| SGBS | | 0.060% | 40m | 0.562% | 4m | 11.550% | 54m | 26.035% | 7.4h |
| EAS | | 0.057% | 6h | 0.496% | 28m | 17.08% | 7.8h | - | - |
| BQ greedy | | 0.579% | 0.6m | 0.895% | 3s | 1.834% | 0.4m | 3.965% | 2.4m |
| BQ bs16 | | 0.046% | 11m | 0.224% | 1m | 0.896% | 6m | 2.605% | 38m |
| LEHD greedy | | 0.577% | 0.4m | 0.859% | 3s | 1.560% | 0.3m | 3.168% | 1.6m |
| LEHD RRC | 50 | 0.0284% | 7.4m | 0.123% | 0.6m | 0.482% | 3.4m | 1.416% | 22m |
| | 100 | 0.0114% | 13.7m | 0.0761% | 1.2m | 0.343% | 8m | 1.218% | 43m |
| | 300 | 0.0044% | 40m | 0.0363% | 3.3m | 0.223% | 22m | 0.899% | 2.1h |
| | 500 | 0.0025% | 1.1h | 0.0280% | 5.3m | 0.193% | 37m | 0.818% | 3.5h |
| | 1000 | **0.0016%** | 2.2h | **0.0182%** | 10.5m | **0.167%** | 1.2h | **0.719%** | 7h |

| | | CVRP100 | | CVRP200 | | CVRP500 | | CVRP1000 | |
|---|---|---|---|---|---|---|---|---|---|
| LKH3 | | 0.000% | 12h | 0.000% | 2.1h | 0.000% | 5.5h | 0.000% | 7.1h |
| HGS | | -0.533% | 4.5h | -1.126% | 1.4h | -1.794% | 4h | -2.162% | 5.3h |
| OR-Tools | | 6.193% | 2h | 6.894% | 1h | 9.112% | 2.2h | 11.662% | 3h |
| MDAM bs50 | | 2.211% | 25m | 4.304% | 3m | 10.498% | 12m | 27.814% | 47m |
| POMO augx8 | | 0.689% | 1m | 4.866% | 7s | 19.901% | 1m | 128.885% | 10m |
| SGBS | | 0.079% | 40m | 2.581% | 1m | 15.343% | 16m | 136.980% | 2.3h |
| EAS | | **-0.234%** | 15h | 0.640% | 33m | 11.042% | 9.3h | - | - |
| BQ greedy | | 2.993% | 0.7m | 3.527% | 4s | 5.121% | 0.4m | 9.812% | 2.4m |
| BQ bs16 | | 0.611% | 10m | 1.141% | 0.6m | 2.991% | 6m | 7.784% | 39m |
| LEHD greedy | | 3.648% | 0.5m | 3.312% | 3s | 3.178% | 0.3m | 4.912% | 1.6m |
| LEHD RRC | 50 | 0.535% | 7.2m | 0.515% | 0.6m | 0.930% | 8m | 2.814% | 27m |
| | 100 | 0.272% | 17m | 0.217% | 1.1m | 0.546% | 14m | 2.370% | 45m |
| | 300 | 0.029% | 52m | -0.146% | 3.6m | 0.045% | 36m | 1.582% | 2.3h |
| | 500 | -0.044% | 1.4h | -0.246% | 6m | -0.107% | 1h | 1.270% | 4h |
| | 1000 | -0.112% | 2.8h | **-0.383%** | 11.3m | **-0.347%** | 2h | **0.921%** | 8h |

Table 2: Experimental results on TSPLib and CVRPLib. "#" denotes the number of instances in the corresponding set.

| | Size | # | POMO aug×8 | BQ greedy | BQ bs16 | LEHD greedy | LEHD RRC |
|---|---|---|---|---|---|---|---|
| TSPLib | <100 | 6 | 0.792% | 1.076% | 0.505% | 0.976% | **0.481%** |
| | 100-200 | 21 | 2.423% | 2.684% | 1.318% | 2.336% | **0.158%** |
| | 200-500 | 15 | 13.413% | 3.177% | 2.183% | 2.742% | **0.200%** |
| | 500-1k | 6 | 31.678% | 8.311% | 5.521% | 4.049% | **1.310%** |
| | >1k | 22 | 63.810% | 42.566% | 36.730% | 11.267% | **4.088%** |
| | All | 70 | 26.439% | 15.668% | 12.923% | 5.260% | **1.529%** |
| | Set (size) | # | POMO aug×8 | BQ greedy | BQ bs16 | LEHD greedy | LEHD RRC |
| CVRPLib | A (31-79) | 27 | 4.970% | 6.310% | 1.627% | 5.871% | **0.647%** |
| | B (30-77) | 23 | 4.747% | 6.859% | 2.221% | 6.049% | **0.812%** |
| | E (12-100) | 11 | 11.402% | 5.884% | 1.211% | 4.809% | **0.541%** |
| | F (44-134) | 3 | 15.973% | 12.568% | 7.404% | 9.051% | **3.009%** |
| | M (100-199) | 5 | 4.861% | 8.407% | 3.691% | 7.094% | **1.817%** |
| | P (15-100) | 23 | 15.525% | 5.902% | 2.393% | 6.611% | **0.917%** |
| | X (100-1k) | 100 | 21.684% | 12.526% | 9.774% | 12.520% | **3.511%** |
| | All | 192 | 15.450% | 9.692% | 6.153% | 9.465% | **2.253%** |

LKH3 on CVRP100-CVRP500 with a budget of 500 RRC iterations and can achieve an around 1% gap to LKH3 on CVRP1000 with 1000 RCC iterations. To the best of our knowledge, our method is the first purely learning-based NCO method that can outperform LKH3 on CVRP200 and CVRP500.

Table 2 shows the test results on real-world TSPLib [43] and CVRPLib [45] instances with different sizes and distributions. We report the results with greedy inference and 1000 RRC iterations for our proposed LEHD method. It is clear that LEHD has a good greedy inference performance and can significantly outperform POMO and BQ on all instances with 1000 RRC iterations. These results demonstrate the robust generalization ability of LEHD.

## 6.2 Ablation Study

Table 3: Comparsion of POMO and LEHD with the same SL training method.

|  | TSP100 | TSP200 | TSP500 | TSP1000 |
|---|---|---|---|---|
| POMO aug×8 SL | **0.571%** | 3.970% | 20.418% | 34.419% |
| LEHD greedy SL | 0.577% | **0.859%** | **1.560%** | **3.168%** |

**Heavy Decoder vs. Heavy Encoder**  We compare the performance of models with the Heavy Encoder structure (POMO) and the Heavy Decoder structure (LEHD). We also train POMO with SL on TSP100 and report the result in Table 3. According to the results, POMO trained by SL can also reach a good training performance on TSP100 while it still performs poorly on large-scale problems. Therefore, the promising generalization performance of our proposed method does come from the heavy decoder structure rather than the SL training.

Table 4: Effect of reinforcement learning and supervised learning on training the LEHD model with 10K testing instances for TSP50/100 and 128 testing instances for TSP200/500/1000.

|  | TSP50 | TSP100 | TSP200 | TSP500 | TSP1000 |
|---|---|---|---|---|---|
| RL | 5.372% | 7.891% | 12.581% | 25.377% | 46.441% |
| SL | **0.375%** | **0.789%** | **1.545%** | **3.500%** | **6.566%** |

**Supervised Learning vs. Reinforcement Learning**  In Table 4, we conduct a comparison with reinforcement learning and supervised learning for training our proposed LEHD model. Due to the high computational cost, the reinforcement learning method with the POMO strategy cannot converge in a reasonable time. Therefore, we use each method to train a LEHD model separately on TSP50 with the same computational budget. The results in Table 4 confirm that our proposed LEHD model is more suitable to be trained by the supervised learning method.

Table 5: Comparison of POMO and LEHD with random sampling and RRC methods.

|  | (10k instances) TSP100 | | (1k instances) TSP200 | | TSP500 | | TSP1000 | |
|---|---|---|---|---|---|---|---|---|
| Concorde | 0.000% | 34m | 0.000% | 23m | 0.000% | 4h | 0.000% | 61h |
| POMO augx8 | 0.134% | 1m | 1.459% | 0.6m | 21.948% | 8m | 40.551% | 1.1h |
| POMO augx8-sample100 | 0.080% | 1.7h | 1.609% | 1.2h | 37.285% | 16h | 82.015% | 117h |
| POMO augx8-RRC100 | 0.115% | 2m | 1.283% | 1.6m | 11.900% | 11m | 25.378% | 1.2h |
| POMO augx8-RRC1000 | 0.075% | 12m | 0.820% | 8m | 6.753% | 29m | 16.261% | 2.1h |
| LEHD greedy | 0.577% | 0.4m | 0.849% | 0.2m | 1.585% | 2.1m | 3.089% | 13m |
| LEHD RRC    100 | 0.0114% | 13.7m | 0.053% | 6.5m | 0.357% | 58m | 1.195% | 5.3h |
| 1000 | **0.0016%** | 2.2h | **0.015%** | 1h | **0.179%** | 9.5h | **0.735%** | 57h |

**The Effect of RRC**  We also conduct comparisons with POMO with random sampling (POMO-Sample100) as well as POMO with RRC (POMO-RRC100) to clearly present the advantages of LEHD and RRC.

The experimental results are shown in Table 5, and we have the following two main observations:

- **POMO-RRC > POMO-Sample**: The random sampling strategy is inefficient in improving POMO's solution quality, especially for large-scale problems. In contrast, our proposed RRC can efficiently improve POMO's performance at inference time.
- **LEHD-RRC > POMO-RRC**: Our proposed LEHD model structure provides enough model capacity for learning to construct partial solutions for different sizes. It is crucial for the strong generalization ability to large-scale problems.

### 6.3 Comparison with Search-based/Improvement-based Methods

Table 6: Comparison with SO-mixed [7] . The results of the SO-mixed method are directly obtained from the original paper.

|  | (128 instances) | | | | | |
|  | TSP200 | | TSP500 | | TSP1000 | |
| --- | --- | --- | --- | --- | --- | --- |
| Concorde | 0.000% | 3m | 0.000% | 32m | 0.000% | 7.8h |
| SO-mixed | 0.636% | 21m | 2.401% | 32m | 2.800% | 56m |
| LEHD greedy | 0.859% | 3s | 1.560% | 0.3m | 3.168% | 1.6m |
| LEHD RRC100 | 0.0761% | 1.2m | 0.343% | 8m | 1.218% | 43m |

Table 7: Comparison with H-TSP [38]. The results of the H-TSP method are directly obtained from the original paper.

|  | TSP1000 (128 instances) | |
| --- | --- | --- |
| Concorde | 0.000% | 7.8h |
| H-TSP with LKH-3 | 4.06% | 7.5m |
| LEHD greedy | 3.168% | 1.6m |
| LEHD RRC100 | 1.218% | 43m |

Recently, Cheng et al. [7] and Pan et al. [38] propose two learning-based methods which can tackle large-scale TSP instances. However, these two methods are search-based/improvement-based approaches that require powerful solvers like LKH3 or specifically designed heuristics, and can only tackle TSP but not other VRP variants. In this work, we propose a purely learning-based construction method to tackle the large-scale TSP/VRP instances, and hence mainly compare with other construction-based methods in the original paper.

We conduct a comparison with these methods for a more comprehensive experimental study. The results are reported in Table 6 and Table 7. Our proposed method clearly outperforms those methods in terms of both performance and running time. It highlights the effectiveness of our proposed LEHD+RRC as a purely learning-based method to tackle large-scale problems.

## 7 Conclusion, Limitation, and Future Work

**Conclusion** In this work, we propose a novel LEHD model for generalizable constructive neural combinatorial optimization. By leveraging the Light Encoder and Heavy Decoder (LEHD) model structure, the LEHD model achieves a strong and robust generalization ability as a purely learning-based method. We also develop a learn to construct partial solution strategy for efficient model training, and a flexible random solution reconstruction mechanism for online solution improvement with customized computational budgets. Extensive experimental comparisons with other representative methods on both synthetic and real-world instances fully demonstrate the promising generalization ability of our proposed LEHD model. Since the generalization ability is one crucial issue for the current constructive NCO solver, we believe LEHD can provide valuable insights and inspire follow-up works to explore the heavy decoder structure for powerful NCO model design.

**Limitation and Future Work** Although with a strong generalization performance, the current LEHD model can only be properly trained by supervised learning. It could be interesting to develop an efficient reinforcement learning-based method for LEHD model training. Another promising future work is to design a powerful learning-based method for more efficient partial solution reconstruction.

## Acknowledgments and Disclosure of Funding

This work was supported in part by the National Natural Science Foundation of China under Grant 62106096, in part by the Shenzhen Technology Plan under Grant JCYJ20220530113013031, in part by the Characteristic Innovation Project of Colleges and Universities in Guangdong Province under Grant 2022KTSCX110, in part by the Research Grants Council of the Hong Kong Special Administrative Region under Grant CityU 11215622, and in part by the 2023 Graduate Innovation Practice Fund Project of Southern University of Science and Technology.

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

# A   Ablation Study of Normalization

## A.1   LEHD Model

In Table 8, we explore the effects of eliminating normalization from the attention layer in our LEHD model. We train three LEHD models with the same training scheme and training budget, differing solely in the attention layer: one with batch normalization (BN), one with instance normalization (IN), and one without normalization (w/o). Our experimental results demonstrate that the LEHD model without normalization in the attention layer can significantly outperform the other two models with normalization.

Table 8: Effect of normalization for LEHD model.

|     | TSP100 | TSP200 | TSP500 | TSP1000 |
|-----|--------|--------|--------|---------|
| BN  | 0.775% | 1.312% | 3.808% | 12.209% |
| IN  | 0.640% | 1.197% | 34.391% | 222.730% |
| w/o | 0.577% | 0.859% | 1.560% | 3.168% |

## A.2   POMO Model

We also compare the performance of POMO [29] with different types of normalization: one with batch normalization (BN), one with instance normalization (IN), and one without normalization (w/o) in Table 9. We train all three POMO models with the same reinforcement learning method with POMO strategy and training budget ($1,000$ epochs). The results show that different types of normalization have little effect on POMO.

Table 9: Effect of normalization for POMO model.

|     | TSP100 | TSP200 | TSP500 | TSP1000 |
|-----|--------|--------|--------|---------|
| BN  | 1.325% | 5.502% | 27.616% | 41.631% |
| IN  | 1.449% | 5.602% | 27.454% | 41.748% |
| w/o | 1.321% | 4.990% | 28.598% | 45.747% |

The results in Table 9 show that removing normalization from the attention layer has little impact on the model with a heavy encoder and a light decoder. However, the results in Table 8 show that removing normalization from the attention layer has a positive impact on the performance of the LEHD model, but it is not the critical factor for the LEHD model's strong generalization ability since the LEHD model with batch normalization still performs significantly better than POMO and SGBS in the case of TSP1000. Instead, we can conclude that the underlying reason for the model's strong generalization ability lies in the heavy decoder structure.

# B   Efficient Self-Improved Training

LEHD uses SL for training, which raises concerns that the performance of LEHD may be upper bounded by the provided (nearly) optimal solutions and the requirement of optimal solutions for LEHD training may make it hard to solve more complex VRPs. We tackle these two concerns as follows.

## B.1   RRC can break the performance upper limit of the labeled solution

Firstly, the performance of our proposed LEHD model with RRC is not limited by the quality of labeled solutions. To demonstrate this point, we train a new LEHD model with labeled solutions generated by OR-Tools of which the quality is far from the optimal solutions (e.g., with a $6.762\%$ optimal gap). The results are shown in Table 10.

In this experiment, we use OR-Tools to generate suboptimal solutions for $100,000$ CVRP100 instances, and use them to train the LEHD model with only 10 epochs (about $1.25$ hours). Then we compare their performance on the same test set with 10k instances. According to the results,

Table 10: The LEHD model's performance trained with suboptimal label outputted by OR-Tools.

|  |  | CVRP100 | |
| --- | --- | --- | --- |
| HGS |  | 0.000% | 4.5h |
| OR-Tools |  | 6.762% | 2h |
| LEHD | greedy | 10.428% | 0.5m |
|  | RRC 20 | 6.249% | 3m |
|  | RRC 50 | 4.811% | 7m |
|  | RRC 100 | 3.915% | 17m |
|  | RRC 300 | 2.862% | 52m |
|  | RRC 500 | 2.525% | 1.4h |

although LEHD with greedy inference is outperformed by OR-Tools, LEHD+RRC can significantly outperform OR-Tools by a large margin with a reasonable runtime. If more training budget is allowed, the performance of LEHD(+RRC) can be further improved.

This result also demonstrates the importance and effectiveness of RRC in our proposed method.

### B.2 LEHD can be trained by RL without any labeled solution

Table 11: The LEHD model's performance trained by RL method on TSP20.

|  |  | (10k instances) | | | |
| --- | --- | --- | --- | --- | --- |
|  |  | TSP20 | | TSP100 | |
| Concorde |  | 0.000% | 3m | 0% | 34m |
| LEHD-RL | greedy | 0.274% | 2s | 5.463% | 0.4m |
|  | RRC100 | 0.014% | 0.6m | 1.271% | 13m |
|  | RRC200 | 0.006% | 1.2m | 0.983% | 26m |
|  | RRC300 | 0.005% | 1.7m | 0.865% | 40m |
|  | RRC500 | 0.003% | 3m | 0.740% | 1.1h |

Secondly, it is possible to directly train the LEHD model using reinforcement learning that does not require any labeled data. Experimental results can be found in Table 11 where we train the LEHD model by RL on the small-scale TSP20 instances, and then test its generalization performance on TSP100. The training budget is 40 epochs each with $100,000$ instances on TSP20 (roughly 9 hours). According to the results, trained by RL, LEHD can achieve a nearly optimal solution for TSP20, and RRC makes LEHD have a reasonably good generalization performance on TSP100.

One concern for the purely RL training for LEHD is the high computational cost and slow convergence speed, as analyzed in Table 4 in the main paper. We design an efficient training method to tackle this concern as explained in the next point.

### B.3 An efficient self-improved training method for LEHD without any labeled solution

Finally, combining the properties from the above two points, we can design an efficient self-improved training method without any labeled solutions for LEHD that can generalize well to large-scale problems. The three key steps are:

- Train LEHD by reinforcement learning with a reasonable computational budget;
- Use LEHD+RRC to generate good solutions for a set of problem instances;
- Further train LEHD by supervised learning with the solutions generated in step 2.

The results of this new training method (LEHD-SI) can be found in Table 12. For the TSP, we first train LEHD by RL and use LEHD+RRC to generate solutions for $200,000$ TSP100 instances as labels for supervised learning. The training budget is 40 RL epochs and 185 SL epochs, which costs 2.7 days in total. The entire training process on TSP does not require external solvers to generate labeled solutions.

Table 12: Experimental results on TSP. 'LEHD-SI' means that the LEHD model is trained by the self-improved training method.

| | | (10k instances) TSP100 | | TSP200 | | (1k instances) TSP500 | | TSP1000 | |
|---|---|---|---|---|---|---|---|---|---|
| Concorde | | 0.000% | 34m | 0.000% | 23m | 0.000% | 4h | 0.000% | 61h |
| Att-GCN+MCTS* | | 0.037% | 2h | 0.884% | 16m | 2.536% | 47m | 3.223% | 1.7h |
| MDAM bs50 | | 0.388% | 21m | 1.944% | 23m | 9.853% | 1.4h | 20.306% | 5.8h |
| POMO augx8 | | 0.134% | 1m | 1.459% | 0.6m | 21.948% | 8m | 40.551% | 1.1h |
| SGBS | | 0.060% | 40m | 0.516% | 30m | 11.398% | 7.3h | 25.997% | 58h |
| EAS | | 0.057% | 6h | 0.619% | 3.6h | 20.322% | 61h | - | - |
| BQ greedy | | 0.579% | 0.6m | 0.892% | 0.4m | 1.862% | 3m | 3.895% | 19m |
| BQ bs16 | | 0.046% | 11m | 0.221% | 6m | 0.945% | 46m | 2.502% | 4.8h |
| LEHD-SI greedy | | 1.073% | 0.4m | 1.445% | 0.2m | 2.583% | 2.1m | 4.523% | 13m |
| LEHD-SI RRC | 100 | 0.053% | 13.7m | 0.183% | 6.5m | 0.756% | 58m | 2.022% | 5.3h |
| | 200 | 0.028% | 27m | 0.129% | 12m | 0.642% | 2h | 1.721% | 10.5h |
| LEHD greedy | | 0.577% | 0.4m | 0.849% | 0.2m | 1.585% | 2.1m | 3.089% | 13m |
| LEHD RRC | 50 | 0.0284% | 7.4m | 0.110% | 3m | 0.475% | 31m | 1.400% | 3.1h |
| | 100 | 0.0114% | 13.7m | 0.053% | 6.5m | 0.357% | 58m | 1.195% | 5.3h |
| | 1000 | **0.0016%** | 2.2h | **0.015%** | 1h | **0.179%** | 9.5h | **0.735%** | 57h |

According to the results, with such a training method, LEHD+RRC can still obtain good generalization performance on TSP instances with up to $1,000$ nodes with a reasonable inference time. Notably, it can outperform the very strong BQ (with bs16) method trained by supervised learning, especially on large-scale problem instances. The performance of LEHD can be further improved if more computational budget is available.

In summary, LEHD can be efficiently trained without any already labeled (nearly) optimal solutions. Therefore, it is possible to extend LEHD to tackle other practical CO problems where the optimal solution is hard to obtain. We will investigate how to design an even more efficient training method for LEHD as well as apply LEHD to solve other CO problems in future work.

## C    Implementation Details for TSP

### C.1    Problem Setup

The task of solving a TSP instance with $n$ nodes involves finding the shortest loop that visits each node exactly once and eventually returns to the first visited node. We generate TSP instances following the approach in Kool et al. [27], where the coordinates of $n$ nodes are sampled uniformly at random from the unit square.

### C.2    Implementation Details

For a TSP instance $\mathbf{S}$, the node features $(\mathbf{s}_1, \ldots, \mathbf{s}_n)$ are the 2-dimentional coordinates of the $n$ nodes in the graph.

In the original AM decoder, irrelevant nodes are masked during each construction step. In our model, we remove the embeddings of irrelevant nodes from the decoder input. This removal serves the same purpose as masking them in every decoder attention layer but also saves computational resources since the decoder is not required to perform computations related to irrelevant nodes. Consequently, for each construction step, the input node embeddings for the decoder only consist of the starting node embedding, the destination node embedding, and the available node embeddings.

Here is an extended explanation of Equation 2 in the case of TSP. After $L$ attention layers, $\widetilde{H}^{(0)}$ is transformed to $\widetilde{H}^{(L)}$. Each vector $\widetilde{\mathbf{h}}_i^{(L)} \in \mathbb{R}^{d_e}$ ($i \neq 1$ or 2) is transformed into a scalar $u_i$ by applying the linear projection $W_O \in \mathbb{R}^{d_e \times 1}$ (i.e., $u_i = W_O \widetilde{\mathbf{h}}_i^{(L)}$), where $d_e$ is the embedding dimension. Then $\mathbf{p}^t = \mathrm{softmax}(\mathbf{u})$ is calculated.

# D Implementation Details for CVRP

## D.1 Problem Setup

A CVRP instance involves $n$ customer nodes and one depot node, with each customer node $i$ having a specific demand $\delta_i$ that must be fulfilled. We aim to determine a set of sub-tours starting and ending at the depot such that the sum of demand satisfied by each sub-tour is within the capacity constraint $D$ of the vehicle. Given the capacity constraint $D$, the objective is to minimize the total distance of the set of sub-tours. Similarly, following Kool et al. [27], we generate CVRP instances where the coordinates of customer nodes and depot nodes are sampled uniformly from the unit square. The demand $\delta_i$ is sampled uniformly form $\{1, \ldots, 9\}$. The vehicle capacities, denoted by $D$, are 50, 80, 100, and 250 for corresponding values of $N$, which are 100, 200, 500, and 1000, respectively.

Following Kool et al. [28] and Drakulic et al. [12], we define the formation of a feasible solution for CVRP. Rather than treating a visit to the depot as a separate step, we use binary variables to indicate whether a customer node is reached via the depot or another customer node. Specifically, in a feasible solution, a node is assigned a value of 1 if it is reached via the depot and a value of 0 if it is reached through another customer node.

For example, a feasible CVRP solution $\{0, 1, 2, 3, 0, 4, 5, 0, 6, 7, 0\}$ where 0 represents the depot, can be denoted as follows:

$$\begin{bmatrix} 1 & 2 & 3 & 4 & 5 & 6 & 7 \\ 1 & 0 & 0 & 1 & 0 & 1 & 0 \end{bmatrix} \tag{4}$$

In this notation, the first row represents the sequence of visited nodes in the solution, and the second row indicates whether each node is reached via the depot or another customer node.

The purpose of using this notation is to ensure solution alignment. In CVRP instances, solutions with the same number of customer nodes may have varying numbers of sub-tours, leading to potential misalignment. By employing this notation, we can avoid such issues.

## D.2 Implementation details

For CVRP, the node feature $\mathbf{s}_i$ is represented as a 3-dimensional vector, comprising the 2-dimensional coordinates and the demand of node $i$. The demand of the depot is assigned a value of 0. Without loss of generality, we normalize the vehicle capacity $D$ to $\hat{D} = 1$, and the demand $\delta_i$ to $\hat{\delta}_i = \frac{\delta_i}{D}$ [27].

In the decoder, the dynamically changing remaining capacity is added to both the starting node and destination node embeddings, resembling the approach employed in Kwon et al. [29]. Similar to TSP, the irrelevant node embeddings are excluded from the decoder input.

Here is an extended explanation of Equation 2 in the case of CVRP. After $L$ attention layers, $\widetilde{H}^{(0)}$ is transformed to $\widetilde{H}^{(L)}$. Each vector $\widetilde{\mathbf{h}}_i^{(L)} \in \mathbb{R}^{d_e}$ ($i \neq 1$ or 2) is projected to a 2-dimensional vector $\mathbf{u}_i$ using the linear projection $W_O \in \mathbb{R}^{d_e \times 2}$ ( i.e., $\mathbf{u}_i = W_O \widetilde{\mathbf{h}}_i^{(L)}$), where $d_e$ is the embedding dimension. Each $\mathbf{u}_i$ corresponds to two actions associated with the node $i$: either being reached via the depot or another customer node. This relation corresponds to the notation mentioned in equation 4. Subsequently, $\mathbf{u}$ is flattened, and the softmax is utilized to compute the probability associated with each possible action.

# E    Solution Visualizations

Table 2 shows the test results on TSPLib and CVRPLib instances with different sizes and distributions. For TSPLib, we report the results on 2D Euclidean TSP instances with sizes smaller than 5000 (up to 4461). For CVRPLib, we report the results on the 2D Euclidean instances without additional constraints such as time windows.

Figures 3 show the solutions of "pr2392" instance in TSPLib. Figures 4 show the solutions of "X-n1001-k43" instance in CVRPLib. For each figure, panel (a) shows the optimal solution, and panel (b), (c), and (d) shows the solution generated by POMO, BQ, and LEHD, respectively.

## E.1    Solution visualizations of two TSPLib instances

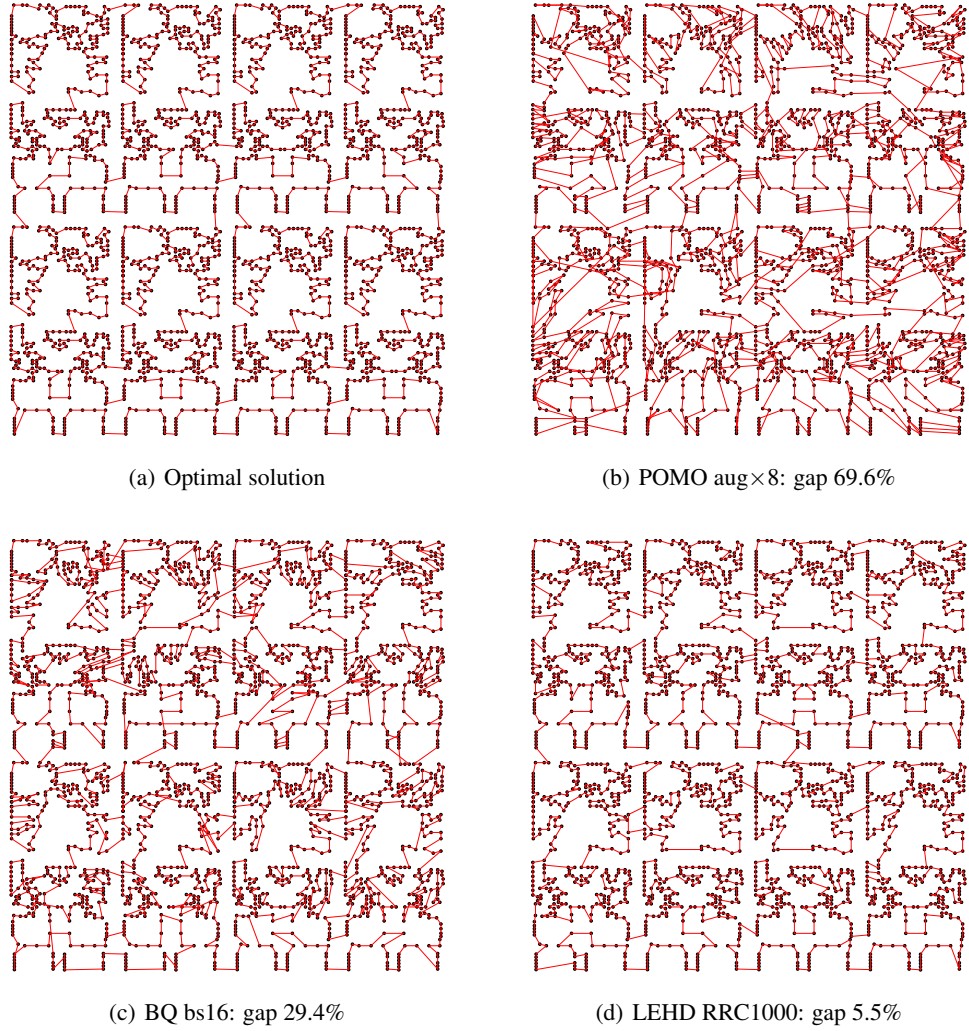

(a) Optimal solution

(b) POMO aug×8: gap 69.6%

(c) BQ bs16: gap 29.4%

(d) LEHD RRC1000: gap 5.5%

Figure 3: Instance pr2392 with 2392 nodes.

## E.2 Solution visualizations of two CVRPLib instances

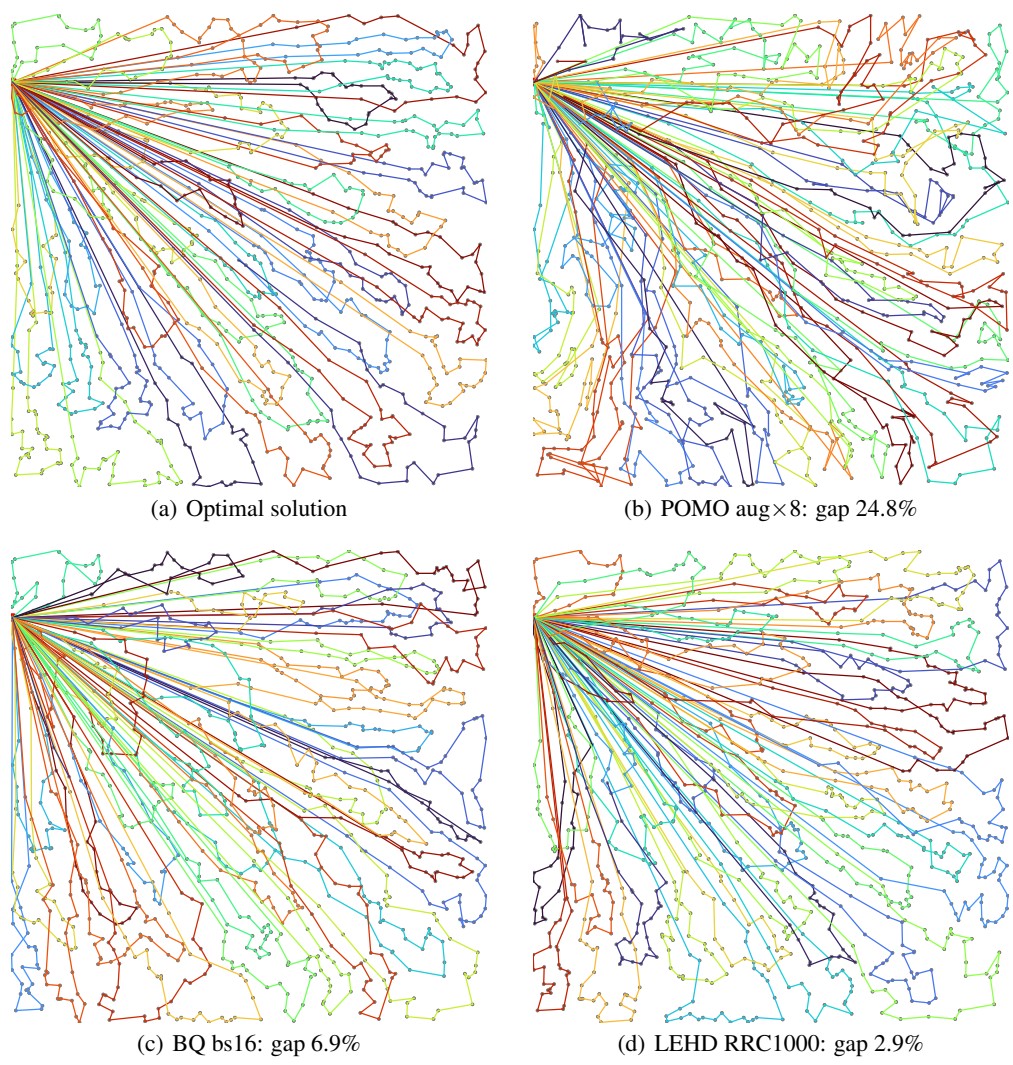

(a) Optimal solution

(b) POMO aug×8: gap 24.8%

(c) BQ bs16: gap 6.9%

(d) LEHD RRC1000: gap 2.9%

Figure 4: Instance X-n1001-k43 with 1000 nodes

# F Licenses

Table 13: List of licenses for the codes and datasets we used in this work

| Resource | Type | Link | License |
|---|---|---|---|
| OR-Tools [41] | Code | https://github.com/google/or-tools | Apache License 2.0 |
| LKH3 [16] | Code | http://webhotel4.ruc.dk/ keld/research/LKH-3/ | Available for academic research use |
| HGS [50] | Code | https://github.com/chkwon/PyHygese | MIT License |
| Concorde [2] | Code | https://github.com/jvkersch/pyconcorde | BSD 3-Clause License |
| POMO [29] | Code | https://github.com/yd-kwon/POMO | MIT License |
| Att-GCN+MCTS [13] | Code | https://github.com/SaneLYX/TSP_Att-GCRN-MCTS | MIT License |
| EAS [19] | Code | https://github.com/ahottung/EAS | Available online |
| SGBS [8] | Code | https://github.com/yd-kwon/SGBS | MIT License |
| TSPLib | Dataset | http://comopt.ifi.uni-heidelberg.de/software/TSPLIB95/ | Available for any non-commercial use |
| CVRPLib | Dataset | http://vrp.galgos.inf.puc-rio.br/index.php/en/ | Available for academic research use |

The licenses for the codes and the datasets used in this work are listed in Table 13.

