# OpenReview forum: "Neural Combinatorial Optimization with Heavy Decoder: Toward Large Scale Generalization"
_NeurIPS.cc/2023/Conference — NeurIPS 2023 poster_

### Official Review · Reviewer_6PbV · 2023-06-22

**Soundness:** 3 good
**Presentation:** 3 good
**Contribution:** 3 good
**Rating:** 5
**Confidence:** 4

**Summary:**

This paper proposes a neural network model in the form of a Light Encoder Heavy Decoder, as well as its training method for solving combinatorial optimization problems. In addition, this paper suggests an RPC method based on updating partial solutions as a way to improve solutions during the inference process. These methods show promising results in experiments with large scale Travelling Salesman Problems (TSP) and Capacitated Vehicle Routing Problems (CVRP).

The approach of learning the state of various node sizes through a heavy decoder is promising. However, using supervised learning as a training method and optaining an optimal solution for training with an existing algorithm has limitations in general use.

**Strengths:**

**S1.** The architecture of Heavy Decoder Model to yield optimal outputs across a range of node sizes

**S2.** Showing promising experimental results on large-scale TSP and CVRP instances

**S3.** Well-written and easy to read

**Weaknesses:**

**W1.** Weak novelty of training method and RPC

**W2.** If there is no optimal solution, or if there is no existing high-performance algorithm such as Concorde/HGS, learning by the method of this paper is not possible, but many practical CO problems cannot secure an optimal solution. Therefore, for most practical CO problems where the optimal solution cannot be known, it is difficult to apply the method in this paper.

**W3.** The optimization degree of the algorithm that outputs the optimal solution (or near optimal solution) of the training data seems to be the upper limit of the optimization degree of LEHD.

**W4.** The number of instances used in the node experiments of 200, 500, and 1000 is too small at 128. It is expected that the experimental values may change as the 128 samples are altered. On the other hand, previous studies [20,8,12] mainly used 1k instances in experiments with 200 or more nodes.
experiments.

**Questions:**

**Q1.** From the experimental results, the inference of the LEHD greedy model was performed faster than the POMO using the HELD model. In the case of TSP 1000, the encoder performs 1 operation and the decoder 1000 operations, so I don't understand how LEHD can be faster than HELD(POMO). If the reason why HELD (POMO) is slower is because of 'augx8', I would like to know the difference in inference time between the HELD(POMO/AM) without augx8 and LEHD.

**Q2.** How to select partial solution to perform update in RPC?

**Q3.** In the CVRP results in Table 1, HGS performed better than LKH3 and was used as the optimal solution for LEHD learning, so it seems better to organize the table with HGS as GAP 0.

**Q4.** Publication information of reference [56] is missed.

---

> ### Author Rebuttal · Authors · 2023-08-09
>
> Thank you very much for your time and effort in reviewing our work. We are glad to know you find our proposed heavy decoder model to learning the state of various node sizes is promising, have promising experimental results on large-scale TSP and CVRP instances, and the paper is well-written.
>
> We address your concerns as follows.
>
> > **W1. Weak novelty of training method and RRC**
>
> Thank you for raising this concern.
>
> The proposed data-efficient training method is important for efficiently learning the heavy decoder model for construction-based NCO. It can be further improved to get rid of the requirement of optimal solutions for efficient training (see the next response).
>
> We have conducted additional experiments to show that  RRC can also improve the POMO performance (see **Q3 for Reviewer 6SQz**), and the construction + RRC approach can outperform some currently proposed divide-and-conquer methods specially designed for TSP (see **Q2 for Reviewer Hd9p**). We will investigate how to further improve RRC (e.g., with learning-based partial solution selection) in future work.
>
> > **W2. Requirement of Optimal Solution**
>
> > **W3. Performance Upper Limit**
>
> Thank you for raising these two important concerns. In short, LEHD with RRC can break the performance upper limit of the labeled solutions, and LEHD can be efficiently trained without any already labeled solutions.
>
> Please see our **General Response (C2)** for a detailed discussion. We will add the new results and discussions to the revised paper.
>
> > **W4. Number of Test Instances**
>
> Thank you for this valuable suggestion. We have now updated Table 1 to report the results with 1k instances for instances with 200 or more nodes. The relative performance among all methods does not change, and the analysis/discussions we have made are still valid.
>
> The revised Table 1 can be found in the **PDF file in General Response**, and will be incorporated into the revised paper.
>
> >**Q1. Inference Time Comparision with POMO**
>
> |                    |   |  TSP100 |      |  TSP200 |       |  TSP500  |      |  TSP1000  |      |
> |--------------------|---|:-------:|:----:|:-------:|:-----:|:--------:|:----:|:---------:|:----:|
> | Concorde           |   | 0.000\% |  34m | 0.000\% |  23m  |  0.000\% |  4h  |  0.000\%  |  61h |
> | POMO single trajec |   | 0.875\% |  3s  | 4.250\% |   2s  | 33.969\% |  5s  |  52.429\% |  11s |
> | POMO no augx8      |   | 0.364\% |  9s  | 2.170\% |   6s  | 23.984\% |  1m  |  42.189\% |  8m  |
> | POMO augx8         |   | 0.134\% |  1m  | 1.459\% |  0.6m | 21.948\% |  8m  |  40.551\% | 1.1h |
> | LEHD greedy        |   | 0.577\% | 0.4m | 0.849\% |  0.2m |  1.585\% | 2.1m |  3.089\%  |  13m |
> |                    |   | CVRP100 |      | CVRP200 |       |  CVRP500 |      |  CVRP1000 |      |
> | HGS                |   | 0.000\% | 4.5h | 0.000\% |  11h  |  0.000\% |  31h |  0.000\%  |  41h |
> | POMO single trajec |   | 3.162\% |  3s  | 9.376\% |   2s  | 35.371\% |  6s  | 306.959\% |  14s |
> | POMO no augx8      |   | 1.216\% |  12s | 6.035\% |   7s  | 30.143\% | 1.2m | 285.635\% |  10m |
> | POMO augx8         |   | 1.229\% |  1m  | 5.192\% |  0.7m | 22.963\% |  9.4m  |  144.230\% | 1.3h |
> | LEHD greedy        |   | 4.170\% | 0.5m | 4.612\% | 0.23m |  5.214\% | 2.2m |  8.594\%  |  13m |
>
> Yes, the reason why HELD (POMO) is slower is because of 'augx8'. We have now included the results of POMO with only a single trajectory (POMO-single-trajec) and regular POMO without augmentation (POMO-no-augx8) for both TSP and CVRP instances in the revised Table 1 which can be found in the **PDF file in General Response**. We also report the runtime comparison in the above table.  According to the results, for TSP1000, the run time for POMO with single trajectory/without augx8/with augx8 are 11s/8m/1.1h, separately, and the run time for LEHD is 13m.
>
> For a problem instance with 1,000 nodes, POMO with a single trajectory only needs to construct 1 path at a time (e.g., like the original AM), POMO without augx8 needs to construct 1,000 paths simultaneously (1,000 different starting nodes), and POMO with augx8 needs to construct 8,000 paths simultaneously. Our proposed LEHD-greedy only needs to greedily construct 1 path at a time, although the decoder requires 1,000 operations. This is the reason why LEHD can be faster than HELD(POMO/AM) aug$\times$8.
>
> We will add the above analysis in the revised paper.
>
> > **Q2. How to select partial solution to perform update in RRC?**
>
> In the current method, we randomly sample the partial solution for the RRC update. The experiment results show that LEHD with simple random RRC can already achieve very promising generalization performance on large-scale TSP/CVRP instances. More advanced learning-based region selection methods could be very helpful to further improve the performance of RRC. We will explore this extension in future work.
>
> > **Q3. HGS as Baseline for CVRP**
>
> Thank you for this good suggestion. We have now updated Table 1 accordingly with HGS as GAP 0.
>
> The revised Table 1 can be found in the **PDF file in General Response**, and will be incorporated into the revised paper.
>
> > **Q4. Publication information of reference [56] is missed.**
>
> Thank you for pointing this out. We have now fixed the publication information of reference [56] as:
>
> [56] Shunyu Yao, Xi Lin, Zhenkun Wang, and Qingfu Zhang. Data-efficient supervised learning is powerful for neural combinatorial optimization. ICLR 2023 Submission. URL: openreview.net/forum?id=a\_yFkJ4-uEK.

---

> > ### Comment · Reviewer_6PbV · 2023-08-14
> > **I will increase the rating of this paper.**
> >
> > My main concerns were W2(Requirement of Optimal Solution) and W3(Performance Upper Limit), I highly appreciate the authors showcasing the potential for addressing these issues through appropriate experiments and additional discussions in General Response C2. I will increase the rating of this paper by 1. I have no further questions. Thank you for the detailed responses and additional experiments.

---

> > > ### Author Response · Authors · 2023-08-14
> > > **Thank you very much for increasing your rating**
> > >
> > > Thank you very much for increasing your rating to 5, and we are glad to know you highly appreciate our response with appropriate experiments and discussions to address your main concerns.
> > >
> > > We would like to know if there is any concern left that makes it hard for you to further increase the rating. We are more than happy to further address any potential concerns.

---

> > > > ### Comment · Reviewer_6PbV · 2023-08-16
> > > >
> > > > Dear authors,
> > > >
> > > > To elaborate further, I believe that a model with the LEHD structure can, of course, be trained using RL. However, when training LEHD with RL, I anticipate there might be challenges compared to when using HELD structure. In the case of the CO problem, there's one Encoder operation followed by multiple Decoder operations (typically over 100 times). And RL generally requires more iterations than SL during training, so a Heavy Decoder structure would likely demand a significantly high computational load for training. Conducting specific experiments on this matter and proposing solutions if issues arise would make this a remarkable study. Additionally, a comparative analysis of accuracy over training time with existing models (like AM/POMO and their subsequent studies) would also be beneficial. I believe the motivation and advantages of LEHD have been well explained in this paper.
> > > >
> > > > While I acknowledge that the authors have demonstrated the potential to address my primary concerns, W2 and W3, in their rebuttal, fundamentally, this paper employs a learning method based on SL. Therefore, I believe a detailed discussion on RL-based training for the LEHD structure model might fall outside the scope of this submitted paper. This is the reason I refrained from asking further questions and limited my rating to 5 points. Nonetheless, I find the approach based on the Heavy Decoder for CO solver to be very impressive and intriguing.

---

> > > > > ### Author Response · Authors · 2023-08-18
> > > > > **Sincerely thank you for your valuable and constructive comments**
> > > > >
> > > > > Sincerely thank you for your valuable and constructive comments. We fully understand your concerns about the efficiency of RL training for LEHD, and we want to provide the following responses:
> > > > >
> > > > > **1. RL training is also extremely inefficient for the classical HELD model on large-scale problems, and it will not hurt our main contribution.**
> > > > >
> > > > > Although most existing constructive HELD models are trained with RL, the problem sizes these methods can properly handle for TSP/CVRP are typically up to 100 (e.g., AM/POMO and follow-up works such as sym-NCO). It could be extremely hard to use RL to train HELD models on problems with a larger size (e.g., 200), while its generalization performance to large-scale problems is very poor. Therefore, the ability to handle large-scale problems is a serious limitation of the popular HELD models.
> > > > >
> > > > > Our proposed LEHD model, even considering the original SL training, is already a crucial contribution for constructive NCO to handle large-scale problems. As shown in the below tables, our proposed RL+SI for LEHD training method is much more efficient than the RL training for classical constructive NCO on CVRP100, while it has significantly better generalization performance. We will carefully incorporate all the related discussions of RL training with more experimental analyses into our revised paper.
> > > > >
> > > > > We totally agree with you that efficient RL training for both HELD and LEHD models is a very important research topic, but we believe it will not hurt our key contribution for promising large-scale generalization in this work.
> > > > >
> > > > >
> > > > > Table A: Training time on CVRP (days)
> > > > > |      Model      | POMO | Sym-NCO |  BQ | LEHD | LEHD-SI |
> > > > > |:---------------:|:----:|:-------:|:---:|:----:|:-------:|
> > > > > | Training method |  RL  |    RL   |  SL |  SL  |  RL+SI  |
> > > > > |       Time      | 13.6 |   11.1  | 1.6 |  2.0 |   2.5   |
> > > > >
> > > > > Table B: Comparison of POMO, Sym-NCO, BQ and LEHD on CVRP.
> > > > > |                  |   |       CVRP100      |      |       CVRP200      |       |       CVRP500      |      |      CVRP1000      |      |
> > > > > |------------------|---|:------------------:|:----:|:------------------:|:-----:|:------------------:|:----:|:------------------:|:----:|
> > > > > | HGS              |   |       0.00\%       | 4.5h |       0.00\%       |  11h  |       0.00\%       |  31h |       0.00\%       |  41h |
> > > > > |  POMO augx8      |   |       1.23\%       |  1m  |       5.19\%       |  0.7m |       22.96\%      | 9.4m |      144.23\%      | 1.3h |
> > > > > | Sym-NCO augx8    |   |       1.45\%       | 1.2m |       6.08\%       |  0.8m |       17.61\%      |  11m |      147.18\%      | 1.6h |
> > > > > | BQ bs16          |   |       1.15\%       |  10m |       2.23\%       |   5m  |       4.98\%       |  50m |       10.39\%      | 5.5h |
> > > > > | LEHD-SI greedy   |   |       5.27\%       | 0.5m |       5.49\%       | 0.23m |       5.92\%       | 2.2m |       9.23\%       |  13m |
> > > > > | LEHD-SI RRC200   |   | $\underline{1.12\\%}$ |  34m | $\underline{1.73\\%}$ |  13m  | $\underline{2.85\\%}$ | 2.1h | $\underline{5.12\\%}$ |  12h |
> > > > > | LEHD greedy      |   |       4.17\%       | 0.5m |       4.61\%       | 0.23m |       5.21\%       | 2.2m |       8.59\%       |  13m |
> > > > > | LEHD RRC300      |   |  $\textbf{0.57\\%}$  |  52m |   $\textbf{0.95\\%}$  |  19m  |   $\textbf{1.94\\%}$  |  3h  |   $\textbf{4.33\\%}$  |  18h |
> > > > >
> > > > >
> > > > > **2. The Contribution of RRC**
> > > > >
> > > > > Furthermore, we do hope that you could re-examine the novelty and contribution of RRC. Although the RRC design is straightforward, it plays a crucial role to achieve promising generalization performance, and is the key to tackle your main concern W3(Performance Upper Limit) and W2(Requirement of Optimal Solution). Indeed, according to the results in Table 1, RRC significantly outperforms all the widely-used (random sampling, beam search) and state-of-the-art advanced search methods (EAS [1], SGBS [2]) for NCO to tackle large-scale generalization. We believe RRC is also an important contribution to the community, since the search method itself is already an important research topic for NCO [1,2].
> > > > >
> > > > > We really appreciate that you think our proposed Heavy Decoder for CO solver to be very impressive and intriguing. We sincerely hope our response can further address your concerns, and your rating is very important to our work.
> > > > >
> > > > > [1] Efficient Active Search for Combinatorial Optimization Problems. ICLR 2022.
> > > > >
> > > > > [2] Simulation-guided beam search for neural combinatorial optimization. NeurIPS 2022.

---

> > > > > > ### Comment · Reviewer_6PbV · 2023-08-18
> > > > > >
> > > > > > I appreciate the additional responses from the authors. My evaluation of the submitted paper remains a lukewarm accept rating of 5. Considering the submitted paper and the discussions so far, I do not see a need to adjust the scoring further.
> > > > > >
> > > > > > The final decision on the submitted paper will be determined by comprehensively considering the content of the paper, the reviewer's evaluation, rebuttal, and discussion. I believe the Area Chair will make an appropriate decision.

---

> > > > > > > ### Author Response · Authors · 2023-08-20
> > > > > > > **Thank you for your comments.**
> > > > > > >
> > > > > > > Thank you again for all your comments. We appreciate your time and effort in engaging with us in the author-reviewer discussion.

---

### Official Review · Reviewer_8Nbk · 2023-06-25

**Soundness:** 4 excellent
**Presentation:** 3 good
**Contribution:** 3 good
**Rating:** 6
**Confidence:** 3

**Summary:**

The paper proposes a learning-based approach called the Light Encoder and Heavy Decoder (LEHD) to solve combinatorial optimization (CO) problems. The proposed method learns less from the static graphs (via a light encoder), and pays more attention to dynamically capturing the relationships between all available nodes of varying sizes (via a heavy decoder), which allows for better generalization to various scales. The devised paradigm naturally divides the problem into sub-problems, and the pipeline is designed to first predict the general route greedily, followed by refining the route using its local construction policy. Experimental evaluations demonstrate the state-of-the-art performance of the proposed method, particularly in terms of its generalization capability.

**Strengths:**

1.The methodology design is insightful and logical. The division into the shortest Hamilton path problem is meaningful for addressing routing problems such as the Traveling Salesman Problem (TSP) and Vehicle Routing Problem (VRP), as it facilitates local improvement naturally.

2.The empirical results are promising especially for the generalization ability.

3.The paper is well-structured and easy to follow.

**Weaknesses:**

1.The RCC algorithm bears resemblance to methods that employ a divide-and-conquer strategy, such as [1] and [2]. It is recommended for the authors to explore and discuss this line of research.

[1] Generalize Learned Heuristics to Solve Large-scale Vehicle Routing Problems in Real-time. ICLR 2023.

[2] Learning to Delegate for Large-scale Vehicle Routing. NeurIPS 2021.

2.The RCC algorithm does not learn to select regions for local improvement, instead randomly enforcing the local policy, which hampers its efficiency.

Minor:

1.Line 166: refind -> refine.
2.Line 224: construction-base -> construction-based.

**Questions:**

1.Why is LKH not included in Table 1? LKH is expected able to achieve optimality for the problem scales in Table 1, but the runtime comparison is essential and should not be missed.

Please also see the discussion of weaknesses.

**Limitations:**

Please see above.

---

> ### Author Rebuttal · Authors · 2023-08-09
>
> Thank you very much for your time and effort in reviewing our work. We are glad to know you find our proposed method design is insightful and logical, the empirical results are promising especially for the generalization ability, and the paper is well-structured and easy to follow.
>
> We address your concerns as follows.
>
> > **W1. RRC and Other Divide-and-Conquer Methods**
>
> Thank you for pointing out these related works.
>
> Divide-and-conquer strategy is indeed a principled way to deal with large-scale methods. These existing works focus on learning how to decompose a large CVRP instance into multiple smaller subproblems, and then use specific and carefully designed solvers (e.g., LKH3) to tackle each subproblems. Since these two methods are specifically designed for CVRP, they cannot be used for TSP and other variants. In contrast, our proposed LEHD is an end-to-end and purely learning-based construction method.
>
> We are currently unable to reproduce the results for [1,2], but we have shown that LEHD can significantly outperform other learning-based methods on large-scale TSP (please see our response to **Q2 for Reviewer Hd9p**). In addition, we have also conducted a new experiment to compare POMO-RRC and LEHD-RRC. The results confirm that while RRC can also efficiently improve POMO's performance, the LEHD model structure itself is crucial for the promising generalization performance on large-scale TSP/CVRP instances (please see our response to **Q3 for Reviewer 6SQz**).
>
> We will add a discussion with [1,2] in the revised paper, and report the comparison results with analysis in the experiment section.
>
> [1] Generalize Learned Heuristics to Solve Large-scale Vehicle Routing Problems in Real-time. ICLR 2023.
>
> [2] Learning to Delegate for Large-scale Vehicle Routing. NeurIPS 2021.
>
> > **W2. Random Selection for RRC**
>
> Totally agree. In this work, we have shown that our proposed LEHD model with simple random RRC can achieve very promising generalization performance on large-scale TSP/CVRP instances. More advanced learning-based region selection methods (such as those in [1,2]) could be very helpful to further improve the performance of RRC. We will explore this extension in future work.
>
> > **W3. Typos**
>
> Thank you for pointing out these typos and they have now been fixed. We will carefully proofread the whole paper to further improve the writing and fix all the typos we can find.
>
> > **Q1. Why is LKH not included in Table 1?**
>
> Thank you for this suggestion, and we have now added the LKH3 results in Table 1 (please see the revised Table 1 in the **PDF file in General Response**). LKH3 can obtain optimal solutions for all TSP instances as expected, and its runtime is a bit longer than the very powerful Concorde solver for TSP.
>
> We will update Table 1 along with more discussion of the new results in the revised paper.

---

> > ### Comment · Reviewer_8Nbk · 2023-08-15
> > **Thanks for the response.**
> >
> > Thanks for your time in the rebuttal. My concerns have been addressed, and I would keep my rating.

---

> > > ### Author Response · Authors · 2023-08-18
> > > **Thank you**
> > >
> > > Glad to know your concerns have been addressed, and thank you for your time and effort in reviewing our work

---

### Official Review · Reviewer_6SQz · 2023-06-30

**Soundness:** 3 good
**Presentation:** 2 fair
**Contribution:** 3 good
**Rating:** 5
**Confidence:** 4

**Summary:**

The paper focuses on a class of NP-hard combinatorial optimization problems called Vehicle Routing Problems (VRP) which is of wide practical interest. This paper proposes a novel Light Encoder and Heavy Decoder (LEHD) model for enabling neural construction methods well handling large-scale routing problems. It incorporates a data-efficient training scheme in the supervised manner and a flexible solution construction mechanism for further enhancing the performance. By training on small-scale problem instances, the LEHD model achieves robust and promising performance on problems with much larger sizes, especially for TSP and CVRP.

**Strengths:**

1. The motivation and training process is clear and easy to read.
2. LEHD could dynamically capture the relationships between all available nodes of varying sizes, which enables the model to generalize well to problems of various scales.
3. Sufficient experiments are conducted to prove the effectiveness and the generalization of LEHD, which has promising results on large-scale TSP and CVRP.

**Weaknesses:**

1. There is some confusion about the training time, since training time is also important.
2. The proposed method might be limited in simple VRPs, since obtaining optimal labeled solutions for more complex VRP variants is hard.
3. The LEHD does not show much superiority to HGS.
4. The presentation of this paper needs to be further improved. Some details are not clearly described and there are some typos.

**Questions:**

1. What is "w" in the experiments? Does it also a random number?
2. For generalizing LEHD to unseen large-scale problems, how does LEHD construct the whole solution?
3. Why not apply "sampling" strategy on POMO with the same iterations of RRC? For example, POMO (sampling) with 100 trials should be compared to LEHD RRC 100.
4. Are the solutions obtained by HGS optimal solutions? To my knowledge, HGS is a meta-heuristic method.



**Limitations:**

Yes

---

> ### Author Rebuttal · Authors · 2023-08-09
>
> Thank you very much for your time and effort in reviewing our work. We are glad to know you find our work has a clear motivation and is easy to read, the proposed LEHD is novel and can generalize well to problems with various sizes, and has sufficient and promising experiment results especially on large-scale TSP and CVRP.
>
> We address your concerns as follows.
>
> > **W1. Training Time**
>
> |      |   | Training time  on CVRP |          |
> |------|:-:|:----------------------:|:--------:|
> | POMO |   |    30500 epochs      | 13.6days |
> | BQ  |   |       1000 epochs      | 1.6 days |
> | LEHD |   |        40 epochs       | 2.0 days |
>
> We totally agree the training time is also important for comparing different methods. Our LEHD model has a comparable running time with the BQ model, while both are much shorter than the POMO training. For CVRP, the training time is roughly 2 days for LEHD, 1.6 days for BQ, and 13.6 days for POMO.
>
> We will add the training time for the learning-based methods in the main paper for a clear comparison.
>
> > **W2. Limitation to Simple VRP**
>
> Thank you for raising this important concern. In short, LEHD can be efficiently trained without any already labeled solutions, and hence it is possible to extend LEHD to tackle complex VRP.
>
> Please see our general response to all reviewers (C2) for a detailed discussion. We will add the new results and discussions to the revised paper.
>
> > **W3. Comparison to HGS**
>
> We agree with the reviewer that LEHD cannot outperform HGS especially on large-scale CVRP instances. However, we want to point out that HGS is a very powerful and currently state-of-the-art heuristic solver for CVRP [1,2], and outperform all the current learning-based solver with a large gap. Indeed, HGS has served as the benchmark algorithm in many competitions [3,4]. However, HGS requires a specific algorithm design with expert knowledge.
>
> Manually designing efficient algorithms to tackle different combinatorial optimization problems (such as complex VRP variants) could be extremely hard. One fundamental motivation for neural combinatorial optimization is to get rid of any handcrafted design to solve CO problems.
>
> As a purely end-to-end learning-based construction method, LEHD has significantly better performance than other neural combinatorial optimization methods. Further experimental studies confirm that LEHD can also significantly outperform other scalable NCO methods that require manually designed solvers for specific problems (see our response to **Q2 for Reviewer Hd9p**). We believe our prosed LEHD model can serve as a strong neural combinatorial optimization baseline for large-scale TSP/CVRP, and inspire more follow-up works from the research community.
>
> [1] A hybrid genetic algorithm for multidepot and periodic vehicle routing problems. Operations Research 2012.
>
> [2] Hybrid genetic search for the CVRP: Open-source implementation and SWAP* neighborhood. Computers \& Operations Research 2021.
>
> [3] DIMACS VRPTW challenge, 2021
>
> [4] EURO Meets NeurIPS Vehicle Routing Competition, NeurIPS 2022
>
> > **W4. Presentations and Typos**
>
> Thank you for pointing this out. We will carefully revise our paper to improve the presentation and fix all the typos. More details of the LEHD model, the training algorithm, and the RRC method will be provided in the main paper.
>
> > **Q1. What is "w" in the experiments? Does it also a random number?**
>
> Yes, "w" is a random number that represents the length of the partial solution that will be constructed by the LEHD model at each step. We uniformly sample $w \sim \textbf{Unif}([4, |V|])$ where $|V|$ is the number of nodes for each TSP/CVRP instance in all experiments.
>
> > **Q2.  For generalizing LEHD to unseen large-scale problems, how does LEHD construct the whole solution?**
>
> Our LEHD model employs a greedy step-by-step approach to construct the whole solution for the given instance $\mathbf{S}$.
>
> In the beginning, a random node is used as both the starting and destination point, and the remaining nodes constitute the available nodes that the model can select step-by-step. Once a node is selected, it will be treated as the new starting node and removed from the set of available nodes, i.e., the starting node is dynamically shifting while the destination node remains constant. When no more available nodes exist, the whole solution is generated. In this way, LEHD can easily construct solutions for problems of various sizes as well as unseen large-scale problems. Once a solution is constructed, the RRC method can be applied to further enhance the solution's quality.
>
> We will carefully revise the training algorithm subsection and add more training details in the experiment section.
>
> > **Q3. Why not apply "sampling" strategy on POMO with the same iterations of RRC?**
>
> Thank you for this valuable suggestion. We have now conducted comparison with POMO with random sampling (POMO-Sample 100) as well as POMO with RRC (POMO-RRC100) to clearly present the advantages of LEHD and RRC.
>
> The experimental results are shown in Table 1 (please see the **PDF file in General Response**), and we have the following two main observations:
>
> - **POMO-RRC > POMO-Sample:** The random sampling strategy is inefficient in improving POMO's solution quality, especially for large-scale problems. In contrast, our proposed RRC can also efficiently improve POMO's performance at inferent time.
>
> - **LEHD-RRC > POMO-RRC:** Our proposed LEHD model structure provides enough model capacity for learning to construct partial solutions for different sizes. It is crucial for the strong generalization ability to large-scale problems.
>
> We have now added the POMO augx8 (sampling 100) results into the revised Table 1. The revised Table 1 and the ablation study (LEHD-RRC v.s. POMO-RRC) will both be added to the revised paper.
>
> > **Q4. Solution Quality of HGS**
>
> HGS is a powerful and currently state-of-the-art heuristic solver for CVRP, please see the above detailed response to W3.

---

> > ### Author Response · Authors · 2023-08-20
> >
> > Thank you again for your time and effort in reviewing our work. Since there is only 1 day left to the discussion deadline, we could want to know whether all of your concerns have been properly addressed. We are more than happy to further address any remaining concerns or questions you might have.

---

### Official Review · Reviewer_Hd9p · 2023-07-07

**Soundness:** 3 good
**Presentation:** 3 good
**Contribution:** 3 good
**Rating:** 7
**Confidence:** 4

**Summary:**

In summary, this work proposed a new solution that generalizes well on large-scale problem instances for routing problems. A data-efficient training scheme and solution construction mechanism are further proposed for efficient training. Experiments verify the effectiveness of the proposed method, and an impressive gap is observed between the proposed method and the existing works.

**Strengths:**

1. The presentation of this manuscript was great, and I enjoy reading this paper.

2. This work proposed a simple yet effective strategy to solve routing problems, which is able to generate nearly optimal solutions for large-scale settings. It also generalized well on real-world datasets. In addition, the effectiveness of such a design is well studied in Section 6.2. I  believe this work would be very insightful for the community.

**Weaknesses:**

1. The training details need more illustration. How did you train on small-scale problem instances? Is the "small-scale problem instances" means the "argument data" you generated?

**Questions:**

1. How did you train on small-scale problem instances? Is the "small-scale problem instances" means the "argument data" you generated?

2. There are some existing works that aim to tackle large-scale TSP instances, for example, [40] and [7] mentioned in your paper. Why no comparison with these works?

**Limitations:**

N.A.

---

> ### Author Rebuttal · Authors · 2023-08-09
>
> Thank you very much for your time and effort in reviewing our work. We are glad to know you enjoy reading our paper, find our method simple yet effective to tackle large-scale routing problems, and believe this work would be very insightful for the community.
>
> We address your concerns as follows.
>
> > **W1/Q1. Training Details:** The training details need more illustration. How did you train on small-scale problem instances? Is the "small-scale problem instances" means the "argument data" you generated?
>
> Thank you for raising this concern. We make the following clarification:
>
> **Small-Scale Problem:** We call a problem with no more than 100 nodes a small-scale problem (e.g., TSP50/100 and CVRP50/100), and a problem with at least 500 nodes a large-scale problem (e.g., TSP500/1000, CVRP500/1000).
>
> **Small-Scale Problem Instances:** Each problem (e.g., TSP100) can have infinite different instances with different node distributions, where each one is a concrete graph with a specific number (e.g., 100) of nodes. Each instance is a data point we consider for model training or inference. In the experiment, we randomly generate 1 million TSP100 instances and 1 million CVRP100 instances to train the LEHD model for TSP and CVRP separately.
>
> **Argumented Data from Partial Solution:** For a routing problem instance $\mathbf{S}$, its optimal solution can be represented by a specific node sequence $\mathbf{x}$ (also called the tour). A partial solution $\mathbf{x}\_{sub}$ is a consecutive subsequence of $\mathbf{x}$. We call the nodes corresponding to $\mathbf{x}\_{sub}$ the augmented data of the original instance $\mathbf{S}$.  For example, a TSP10 instance $\mathbf{S} \subset \mathbb{R}^{2\times 10}$ consists of $10$ nodes, and each column is the coordinates of the node. Its optimal solution $\mathbf{x}$ is a permutation of the ten nodes, e.g., $\mathbf{x}=[8,7,9,1,4,5,3,6,2,0]$. The partial solution $\mathbf{x}\_{sub}$ is a consecutive subsequence sampled from $\mathbf{x}$ with a random size and direction. For example, $\mathbf{x}\_{sub}$ can be $[6,3,5,4,1]$ or $[1,4,5,3,6]$ if its size is five. The partial solution $\mathbf{x}_{sub}$ along with its correponding nodes is one augmented data point in our method.
>
> **Model Training with Partial Solutions:** The model learns to construct the partial solution node by node. The first node of $\mathbf{x}\_{sub}$ is the starting node, the last one of $\mathbf{x}\_{sub}$ is the destination node, and the remaining nodes constitute the available nodes to be selected by the model. For the example $\mathbf{x}\_{sub}=[6,3,5,4,1]$, node $6$ is the starting node, node $1$ is the destination node, and nodes $3,5,4$ are available nodes.
>
> In each training step, the model provides each available node $s\_i$ with a probability $p\_i$ of being selected. Meanwhile, the label $\mathbf{x}\_{sub}$ tells whether the node $s_i$ should be chosen in the current step (i.e., $y\_i = 1$ or $0$). The cross-entropy loss $loss = -\sum\_{i=1}^{u} y\_i \log(p\_i)$ is employed. Then the Adam optimizer computes gradients based on the loss and updates the model's parameters. Throughout the training process, the number of available nodes gradually decreases, and the starting node dynamically shifts to the node selected in the previous step while the destination node remains constant. The training epoch ends when there are no more nodes available. The next epoch starts with new augmented data.
>
> This data-efficient training method enables our powerful LEHD model to be trained on small-scale problem instances and then generalize well to much larger problem instances. We will carefully revise the algorithm section to make the training details clear and easy to follow, and also add an intuitive example in the appendix for better illustration.
>
> > **Q2. Other Works for Large-Scale TSP:** There are some existing works that aim to tackle large-scale TSP instances, for example, [40] and [7] mentioned in your paper. Why no comparison with these works?
>
> |               |   | TSP1000 (128 instances) |      |
> |---------------|---|:-----------------------:|:----:|
> | Concorde      |   |         0.000\%         | 7.8h |
> | H-TSP with LKH3        |   |          4.06\%         |  7.5m |
> | LEHD greedy   |   |         3.168\%         | 1.6m |
> | LEHD (RRC100) |   |         1.218\%         |  43m |
>
> |               |   | (128 instances) |      |         |      |         |      |
> |---------------|---|:---------------:|:----:|:-------:|:----:|:-------:|:----:|
> |               |   |      TSP200     |      |  TSP500 |      | TSP1000 |      |
> | Concorde      |   |   0.000\% | 3m| 0.000\% |  32m | 0.000\% | 7.8h |
> | SO-mixed      |   |   0.636\%  | 21m | 2.401\% |  32m | 2.800\% |  56m |
> | LEHD greedy   |   |   0.859\%  | 3s  | 1.560\% | 0.3m | 3.168\% | 1.6m |
> | LEHD (RRC100) |   |   0.0761\% |1.2m | 0.343\% |  8m  | 1.218\% |  43m |
>
> Thank you for this valuable suggestion.
>
> The existing works [7,40] are search-based/improvement-based methods that require powerful solvers like LKH3 or specifically designed heuristics, and can only tackle TSP but not other VRP variants. In this work, we propose a purely learning-based construction method to tackle the large-scale TSP/VRP instances, and hence mainly compare with other construction-based methods in the original paper.
>
> Following your suggestion, we will add a comparison with these methods for a more comprehensive experimental study. Some preliminary results are reported in the above table. It is clear that our proposed method can outperform those methods both in performance and running time. It highlights the effectiveness and usefulness of our proposed LEHD+RRC as a purely learning-based method to tackle large-scale problems.
>
> [7] Select and optimize: Learning to solve large-scale tsp instances. AISTATS 2023.
>
> [40] H-TSP: Hierarchically solving the large-scale traveling salesman problem. AAAI 2023.

---

> > ### Comment · Reviewer_Hd9p · 2023-08-18
> > **Thanks for the response.**
> >
> > My previous concerns are well addressed. Thanks for the detailed clarification and experimental results.
> > I have one additional question, in Table 1, the POMO augx8 gets 40.57% on TSP1000 and 128.88% on CVRP1000, and I see in Table 4, LEHD with RL gets 46.441% on TSP1000. Given the 6.56% achieved by the RL training approach, is the additional gain between your method and POMO comes from the supervised training process rather than the heavy decoder design?
> > It would be great to conduct a discussion of why the RL-based methods (e.g., POMO and LEHD model with RL) do not generalize well on large scaled problems in Section 6.1 or 6.2.

---

> > > ### Author Response · Authors · 2023-08-20
> > > **Thank you very much for your comment [1/2].**
> > >
> > > Thank you very much for your comment and glad to know your previous concerns are well addressed. We address your new question as follows.
> > >
> > > > **1. The poor generalization performances of LEHD-RL and POMO are caused by completely different reasons.**
> > >
> > > **LEHD-RL:** The poor performance of LEHD-RL comes from the extremely inefficient RL training for LEHD. Our original goal of Table 4 is to compare the RL v.s SL training for our proposed  LEHD model. In this experiment, all methods are trained on TSP50 (rather than TSP100 in Table 1) with a limited training budget (2 days rather than 2 weeks for POMO in Table 1). According to the result, we find that LEHD-SL can reach a good training performance on TSP50 ($0.375\\%$) while LEHD-RL cannot converge and has a poor training performance on TSP50 ($5.372\\%$). The poor training performance of LEHD-RL directly leads to its poor large-scale generalization performance.
> > >
> > > Table A: Effect of RL and SL on training the LEHD model with 10K testing instances for TSP50/100 and 1K testing instances for TSP200/500/1000.
> > > |              |       TSP50      |      TSP100      |      TSP200      |      TSP500      |      TSP1000     |
> > > |--------------|:----------------:|:----------------:|:----------------:|:----------------:|:----------------:|
> > > | RL (2.2days) |      5.372\%     |      7.891\%     |     13.379\%     |     25.685\%     |     46.463\%     |
> > > | RL (6.9days) |      2.576\%     |      4.390\%     |      9.398\%     |     22.318\%     |     40.911\%     |
> > > | SL (2.2days) | $\textbf{0.375\%}$ | $\textbf{0.789\%}$ | $\textbf{1.502\%}$ | $\textbf{3.528\%}$ | $\textbf{6.623\%}$ |
> > >
> > > In the preliminary study of this project, we have tried to train LEHD-RL longer to check its performance, and the result is reported in the above table. With more computational budget (e.g., 6.9 days), LEHD-RL can obtain better training performance on TSP50 and hence also better generalization performance on large-scale problems. However, the RL training for LEHD still requires a much longer time to converge. This high computational cost for LEHD-RL is indeed expected since the heavy decoder needs to be called $n$ times at each step for training on each instance with size $n$. We will add this result and discussion on RL training into the revised paper, along with a novel self-improvement RL method to tackle this issue (see point 3 below).
> > >
> > > **POMO:** The poor performance of POMO on large-scale problems actually comes from its poor generalization ability. In Table 1, we train POMO with RL on TSP100 with a large computational budget (e.g., 2 weeks). POMO has a quite promising training performance on TSP100 (e.g., $0.134\\%$ with augx8) in Table 1, which means it is sufficiently trained on TSP100. However, its generalization performance dramatically decreases and obtains a $40.551\\%$ gap on TSP1000. This poor generalization performance of POMO (and other HELD models) has also been observed in many existing NCO works, and the supervised training process cannot overcome such a generalization gap (see next point).
> > >
> > > > **2. The heavy decoder design is crucial for the promising generalization performance.**
> > >
> > > To investigate the effect of model structure on the generalization performance, we also train POMO with SL on TSP100 and report the result in the table below. According to the results, POMO trained by SL can also reach a good training performance on TSP100 while it still performs poorly on large-scale problems. Therefore, the promising generalization performance of our proposed method does come from the heavy decoder structure rather than the SL training.
> > >
> > > Table B: POMO-SL v.s. LEHD-SL
> > >
> > > |                |      TSP100      |      TSP200      |       TSP500      |      TSP1000     |
> > > |----------------|:----------------:|:----------------:|:-----------------:|:----------------:|
> > > | POMO aug×8 SL  | **0.571\%** |      3.970\%     |      20.418\%     |     34.419\%     |
> > > | LEHD greedy SL |      0.577\%     | **0.859\%** | **1.560\%** | **3.168\%** |
> > >
> > > **Why the heavy decoder model is better for generalization:**
> > >
> > > As discussed in the main paper, the current HELD model aims to learn the embeddings of all nodes via a heavy encoder in one shot and then sequentially construct the solution with the static node embeddings via a light decoder. This one-shot embedding learning may incline the model to learn scale-related features to perform well on training instances, but hinder the model from capturing the necessary relations among a significantly larger number of nodes.
> > >
> > > In contrast, our proposed LEHD model learns to dynamically capture the relationships among the current partial solution and all the available nodes at each construction step via the heavy decoder. As the size of nodes varies with construction steps, the model tends to learn scale-independent features. Therefore, it could be less sensitive to the instance size and has much better generalization performance on large-scale problem instances.

---

> > > > ### Author Response · Authors · 2023-08-20
> > > > **Thank you very much for your comment [2/2].**
> > > >
> > > > > **3. The shortcomings of both SL/RL training for LEHD can be overcome by RL-SI.**
> > > >
> > > > As in the **General Response C2**, we have proposed a novel self-improved method with reinforcement learning (RL-SI) to overcome the crucial shortcoming for both SL (requirement of labeled optimal solution) and RL (high computational cost) training for LEHD. As shown in the below tables, our proposed RL-SI training for LEHD is much more efficient than the RL training for classical constructive models (e.g., POMO and Sym-NCO) on CVRP100, while it has significantly better generalization performance.
> > > >
> > > > Table C: Training time on CVRP100 (days).
> > > > |      Model      | POMO | Sym-NCO | LEHD | LEHD-SI |
> > > > |:---------------:|:----:|:-------:|:----:|:-------:|
> > > > | Training method |  RL  |    RL   |  SL  |  RL+SI  |
> > > > |       Time      | 13.6 |   11.1  |  2.0 |   2.5   |
> > > >
> > > >
> > > > Table D: Comparison of POMO, Sym-NCO and LEHD on CVRP.
> > > > |                  |   |       CVRP100      |      |       CVRP200      |       |       CVRP500      |      |      CVRP1000      |      |
> > > > |------------------|---|:------------------:|:----:|:------------------:|:-----:|:------------------:|:----:|:------------------:|:----:|
> > > > | HGS              |   |       0.00\%       | 4.5h |       0.00\%       |  11h  |       0.00\%       |  31h |       0.00\%       |  41h |
> > > > |  POMO augx8      |   |       1.23\%       |  1m  |       5.19\%       |  0.7m |       22.96\%      | 9.4m |      144.23\%      | 1.3h |
> > > > | Sym-NCO augx8    |   |       1.45\%       | 1.2m |       6.08\%       |  0.8m |       17.61\%      |  11m |      147.18\%      | 1.6h |
> > > > | LEHD-SI greedy   |   |       5.27\%       | 0.5m |       5.49\%       | 0.23m |       5.92\%       | 2.2m |       9.23\%       |  13m |
> > > > | LEHD-SI RRC200   |   | $\underline{1.12\%}$ |  34m | $\underline{1.73\%}$ |  13m  | $\underline{2.85\%}$ | 2.1h | $\underline{5.12\%}$ |  12h |
> > > > | LEHD greedy      |   |       4.17\%       | 0.5m |       4.61\%       | 0.23m |       5.21\%       | 2.2m |       8.59\%       |  13m |
> > > > | LEHD RRC300      |   |   **0.57\%**  |  52m |   **0.95\%**  |  19m  |   **1.94\%**  |  3h  |   **4.33\%**  |  18h |
> > > >
> > > > All the above discussions will be carefully added to the revised paper. Our code and model will also be open-source upon publication.

---

> > > > > ### Comment · Reviewer_Hd9p · 2023-08-20
> > > > >
> > > > > Thanks for the prompt response, I would like to increase my rating to 7 in this case.

---

> > > > > > ### Author Response · Authors · 2023-08-20
> > > > > > **Thank you very much.**
> > > > > >
> > > > > > Thank you very much for your effort in reviewing our paper and engaging with us in the discussion. We are thrilled to know your concerns are all well addressed, and you have increased your rating to 7.

---

### Author Rebuttal · Authors · 2023-08-09

**General Response**

Dear AC and Reviewers,

We sincerely thank you for your time and effort in reviewing our work. We are glad to know the reviewers enjoy reading our paper  or find it easy to follow (**Hd9p**, **6SQz**, **8Nbk**, **6Pbv**), our method is effective (**Hd9p**) /novel (**6SQz**) /insightful (**8Nbk**) /promising (**6PbV**), has promising generalization performance on large-scale problems (**Hd9p, 6SQz, 8Nbk, 6PbV**), and believe this work could be very insightful to the community (**Hd9p**).

We address some common concerns shared by different reviewers in this response.

> **C1. More Experimental Results**

Following your valuable suggestions, we have put more experimental results in Table 1. The revised Table 1 is attached to this comment and will be added to the revised paper.

Detailed discussions of these new results can be found in the responses to each individual reviewer.

> **C2. Will the requirement of optimal solutions for LEHD training make it hard to solve more complex VRPs? Will the performance of LEHD upper bounded by the provided (nearly) optimal solutions?**

For a better discussion, we tackle these two concerns in a reverse order.

**1. RRC can break the performance upper limit of the labeled solution**

Firstly, the performance of our proposed LEHD model with RRC is not limited by the quality of labeled solutions. To demonstrate this point, we train a new LEHD model with labeled solutions generated by OR-Tools of which the quality is far from the optimal solutions (e.g., with a 6.762% optimal gap). The results are shown in the following table:

|           |         | CVRP100  |      |
|-----------|:-------:|:--------:|:----:|
| HGS       |         |  0.000\% | 4.5h |
| OR-Tools  |         |  6.762\% |  2h  |
| LEHD      |  greedy | 10.428\% | 0.5m |
|           |  RRC 20 |  6.249\% |  3m  |
|           |  RRC 50 |  4.811\% |  7m  |
|           | RRC 100 |  3.915\% |  17m |
|           | RRC 300 |  2.862\% |  52m |
|           | RRC 500 |  2.525\% | 1.4h |

In this experiment, we use OR-Tools to generate suboptimal solutions for $100,000$ CVRP100 instances, and use them to train the LEHD model with only 10 epochs (about $1.25$ hours). Then we compare their performance on the same test set with 10k instances. According to the results, although LEHD with greedy inference is outperformed by OR-Tools, LEHD+RRC can significantly outperform OR-Tools. If more training budget is available, the performance of LEHD(+RRC) can be further improved.

This result also demonstrates the effectiveness of RRC in our proposed method.

**2. LEHD can be trained by RL without any labeled solution**

Secondly, it is possible to directly train the LEHD model using reinforcement learning without any labeled data. Experimental results can be found in the following table:

|          |        | (10k instances) |      |         |      |
|----------|:------:|:---------------:|:----:|:-------:|:----:|
|          |        |      TSP20      |      | TSP100  |      |
| Concorde |        |     0.000\%     |  3m  |   0\%   |  34m |
| LEHD-RL  | greedy |     0.274\%     |  2s  | 5.463\% | 0.4m |
|          | RRC100 |     0.014\%     | 0.6m | 1.271\% |  13m |
|          | RRC200 |     0.006\%     | 1.2m | 0.983\% |  26m |
|          | RRC300 |     0.005\%     | 1.7m | 0.865\% |  40m |
|          | RRC500 |     0.003\%     |  3m  | 0.740\% | 1.1h |

Here, we train the LEHD model by RL on the small-scale TSP20 instances, and then test its generalization performance on TSP100. The training budget is 40 epochs each with 100,000 instances on TSP20 (roughly 9 hours). According to the results, LEHD can achieve nearly optimal solution for TSP20, and RRC enables LEHD to have a reasonably good generalization performance on TSP100.

One concern for the purely RL training for LEHD is the high computational cost as analyzed in Table 4 in the main paper. We design an efficient training method to tackle this concern in the next point.

**3. An efficient self-improved training method for LEHD without any labeled solution**

Finally, combing the above two points, we can design an efficient self-improved method without any labeled solutions to train LEHD that can generalize well to large-scale problems. The three key steps are:

- Step 1. Train LEHD by reinforcement learning with a reasonable computational budget;

- Step 2. Use LEHD+RRC to generate good solutions for a set of problem instances;

- Step 3. Further train LEHD by supervised learning with the solutions generated in step 2;

Results of this method (LEHD-SI) can be found in the revised Table 1 (see attachment). For TSP, we first train LEHD by RL and use LEHD+RRC to generate solutions for 200,000 TSP100 instances as labels for supervised learning. The training budget is 40 RL epochs and 185 SL epochs, which costs 2.7 days in total. For CVRP, the training budget is 80 RL epochs and 70 SL epochs, which cost 2.5 days in total. The entire training process on TSP and CVRP does not require external solvers to generate labeled solutions.

According to the results, LEHD-SI-RRC can still obtain good generalization performance on TSP/CVRP instances with up to 1,000 nodes with a reasonable inference time. Notably, it can outperform the very strong BQ (with bs16) method trained by supervised learning, especially on large-scale problem instances. The performance of LEHD-SI can be further improved if more computational budget is available.

**In summary**, LEHD can be efficiently trained without any already labeled (nearly) optimal solutions. Therefore, it is possible to extend LEHD to tackle other practical CO problems where the optimal solution is hard to obtain. We will investigate how to design an even more efficient training method for LEHD as well as apply LEHD to solve other CO problems in future work. All the above discussion will be carefully added into the revised paper.

Best Regards,

Paper8077 Authors

---

> ### Comment · Area_Chair_j6G1 · 2023-08-13
> **lets followup author response**
>
> Hi all,
>
> Thanks for serving as the reviewers for this submission. As the authors have already provided their responses. Now let's start further discussion. Here is a to-do list:
>
> (1) Please acknowledge the authors when you finish reading their responses.
> (2) Please indicate whether you have any further questions for the authors such that they can continue to response.
> (3) Please indicate whether you are willing to change the ratings.
>
> best,
> The AC

---

### Decision · Program_Chairs · 2023-09-21

**Decision:**

Accept (poster)

**Comment:**

The paper leads to an important direction for machine learning for combinatorial optimization (CO): increasing the capacity of the decoder to handle large-scale CO problems. The reviewers found the paper was clearly motivated and well written with convincing experimental results. Also, the technical novelty is good. The authors have addressed the reviewers' questions and all the reviewers gave positive ratings to this paper. Based on these facts and also based on my personal rating, I suggest accepting this paper.